# Structure and assembly of double-headed Sendai virus nucleocapsids

Na Zhang[1,2,3,9], Hong Shan[1,9], Mingdong Liu[1,3,9], Tianhao Li[1,3,9], Rui Luo[4], Liuyan Yang[5,6], Lei Qi[2], Xiaofeng Chu[1,3], Xin Su[1,3], Rui Wang[1], Yunhui Liu[1], Wenzhi Sun [7,8] & Qing-Tao Shen [1,2✉]

Paramyxoviruses, including the mumps virus, measles virus, Nipah virus and Sendai virus (SeV), have non-segmented single-stranded negative-sense RNA genomes which are encapsidated by nucleoproteins into helical nucleocapsids. Here, we reported a double-headed SeV nucleocapsid assembled in a tail-to-tail manner, and resolved its helical stems and clam-shaped joint at the respective resolutions of 2.9 and 3.9 Å, via cryo-electron microscopy. Our structures offer important insights into the mechanism of the helical polymerization, in particular via an unnoticed exchange of a N-terminal hole formed by three loops of nucleoproteins, and unveil the clam-shaped joint in a hyper-closed state for nucleocapsid dimerization. Direct visualization of the loop from the disordered C-terminal tail provides structural evidence that C-terminal tail is correlated to the curvature of nucleocapsid and links nucleocapsid condensation and genome replication and transcription with different assembly forms.

[1] iHuman Institute, School of Life Science and Technology, ShanghaiTech University, Shanghai, China. [2] Laboratory for Marine Biology and Biotechnology, Qingdao National Laboratory for Marine Science and Technology, Qingdao, China. [3] University of Chinese Academy of Sciences, Beijing, China. [4] State Key Laboratory of Agricultural Microbiology, College of Veterinary Medicine, Huazhong Agricultural University, Wuhan, China. [5] State Key Laboratory of Microbial Technology, Marine Biotechnology Research Center, Shandong University, Qingdao, China. [6] College of Marine Life Sciences, Ocean University of China, Qingdao, China. [7] Chinese Institute for Brain Research, Beijing, China. [8] School of Basic Medical Sciences, Capital Medical University, Beijing, China. [9] These authors contributed equally: Na Zhang, Hong Shan, Mingdong Liu, Tianhao Li. ✉email: shenqt@shanghaitech.edu.cn

The family of *Paramyxoviridae* consists of many human viruses such as measles, parainfluenza, and mumps viruses, and animal-derived pathogens including Newcastle disease and Sendai viruses. Paramyxoviruses have single-stranded negative-sense RNA genomes, and their non-segmented viral genomes are encapsulated by many copies of a nucleoprotein (N), as well as other viral proteins, forming long and helical nucleocapsids (NC) that act as scaffolds for virus assembly and as templates for genome transcription and replication[1,2]. During viral RNA synthesis, the viral phosphoprotein recognizes the C-terminal tail (N-tail) of a nucleoprotein to guide RNA polymerase on the nucleocapsid to synthesize daughter RNA[3,4]. The nascent RNA strand is immediately enwrapped by nucleoproteins for protection against possible digestions by nucleases in the host cell, before being packed into virions; this packing is also known to be mediated by nucleoproteins[5].

Despite of poor sequence conservation, paramyxovirus nucleoproteins exhibit well-conserved architectures[2]. Specifically, paramyxovirus nucleoproteins typically consist of two lobes—an N-terminal domain (NTD) and a C-terminal domain (CTD)—with a cleft between the lobes. The interdomain cleft comprises conserved positively charged residues which function to promote clamping to nucleotides of the RNA genome, with no apparent specificity[5]. Both the NTD and CTD have subdomains known as arms (N-arm and C-arm, respectively), which enable paramyxovirus nucleoproteins to undertake a "domain swapping" process that enables their assembly into either ring-like structures in parainfluenza virus 5 (PIV5) or helical filaments in measles virus (MeV)[6–8].

Expanding beyond the known helical nucleocapsids, we recently described a clam-shaped assembly of the nucleoprotein from Newcastle disease virus (NDV), wherein two single-turn spirals are packed in a tail-to-tail way[9]. Each single-turn spiral in NDV clam-shaped assembly is similar to MeV helical nucleocapsid and enwraps one RNA molecule between NTD and CTD in a "3-bases-in, 3-bases-out" conformation. Surprisingly, there is an obvious seam between two single-turn spirals, which disconnects two RNA molecules. The clam-shaped assemblies of NDV nucleoproteins are mediated by loops (residues 114–120) of vertically adjacent nucleoproteins in the clam-shaped core. Intriguingly, this clam-shaped architecture suggested the possibility of acting as a seed to nucleate the further formation of a double-headed type of nucleocapsid[9].

SeV is responsible for a respiratory tract infection among murine rodents via both airborne and direct contact routes, potentially transmissible to humans as many animal-derived pathogens such as SARS, MERS, and COVID-19[10–12]. Similar to PIV5[13], SeV virion is highly pleomorphic and has been shown to range in diameter from ~110 to 540 nm, indicating that some virions may contain multiple copies of their genomes[14]. Furthermore, SeV has been proven infectious to many human cancer cell lines and has been shown to have oncolytic properties in animal models[15,16]. Given the understanding that biomedical technologies based on SeV have great potential for treating various cancers, obtaining detailed information for the SeV nucleoprotein structure and assembly process will both deepen a basic understanding of the underlying mechanism of SeV pathogenesis and guide efforts to develop novel anti-cancer therapies.

## Results

### Structure of double-headed SeV nucleocapsid.
Following previous reports, SeV nucleoproteins were purified from *Escherichia coli* after tandem affinity and gel-filtration chromatography. Similar to NDV and MeV[9,17], SeV nucleoprotein exhibits structural variability from filaments to ring-like structures. The early-eluting fraction at 8.7 mL mainly consists of filaments; all the filaments have a diameter at ~19 nm, and 60% of the nucleocapsids observed are of the double-headed type, wherein two herringbone-like filaments are joined as a clam-shaped structure in a tail-to-tail manner (Supplementary Fig. 1a–d). The reconstruction strategy is to divide double-headed SeV nucleocapsids into helical stems and clam-shaped joints based on the symmetries, reconstruct their respective high-resolution structures, and then splice them together.

The relatively straight helical stems of the double-headed SeV nucleocapsids were selected for our helical reconstruction. After three-dimensional (3D) classification, two conformers were resolved, exhibiting slight differences in the extent of their helical twist and rise, with the respective resolutions of 4.1 and 4.6 Å (Supplementary Figs. 2, 3a and Table 1). Following the previous treatments on mumps virus (MuV) phosphoproteins and nucleoproteins[4,18], the purified SeV nucleoproteins were placed at 4 °C for 5 weeks. This approach limited the cleavage of residual impurities only onto the N-tail (404-524) of nucleoproteins and yielded much straighter filaments (hereafter denoted as $N_{cleaved}$, with the respective nucleocapsid denoted as $NC_{cleaved}$ and the uncleaved nucleoprotein and nucleocapsid denoted as $N_{WT}$ and $NC_{WT}$) (Supplementary Fig. 1e). These $NC_{cleaved}$ samples yielded a much higher resolution nucleocapsid at 2.9 Å (Supplementary Figs. 3b, 4a–c and Supplementary Movie 1).

An atomic model of the SeV nucleoprotein was successfully built covering residues 3 to 414 based on the model of MeV nucleoprotein with 27% sequence identity (Fig. 1b, c and Supplementary Fig. 4d), our SeV nucleoprotein structure was coupled with RNA, without apparent sequence specificity. Poly-Uracil was docked into the $NC_{cleaved}$ EM map to mimic cellular RNA, with both the sugar-phosphate backbones and nitrogenous bases were clearly resolved. (Fig. 1b, c and Supplementary Fig. 4e). Structural comparison between SeV nucleoprotein and other paramyxovirus nucleoproteins such as NDV, MuV, PIV5, MeV, and Nipah virus (NiV)[6–9,19], indicates high structural conservation with the RMSD values <1.3 Å (Supplementary Fig. 5).

The two helical stems were joined as a clam-shaped structure. Similar to NDV[9], there are many dispersed ring-like particles in the late-eluting fraction during the purification of SeV nucleoprotein (Supplementary Fig. 1a–c). Direct 2D classification indicates a strong preferred orientation of dispersed ring-like particles. To increase side views of clam-shaped structures, joints between two helical stems in each filament were manually picked up and combined with dispersed ring-like particles. 3D reconstruction on the merged particle sets yields a clam-shaped structure at 5.6 Å resolution, no matter whether two-fold symmetry is enforced or not (Supplementary Figs. 6 and 7a–c). The top-on view of the clam-shaped structure highlights its crescent shape, in which the protomers further away from the gap are resolved much better than those closer to the gap due to the structural wobbling (Supplementary Fig. 7d). Based on the local symmetry, we employed a strategy to improve resolution by masking and averaging the well-resolved 5 pairs of consecutive protomers; the final cryo-EM map reaches 3.9 Å of the resolution, into which the atomic model of the SeV nucleoprotein got from the above helical stem was fitted well via rigid-body docking (Fig. 1d and Supplementary Figs. 7e, f, 8a, b, Supplementary Movie 2). Clear RNA densities are also evident in the clam-shaped EM map, and poly-Uracil is readily docked into the density.

Interestingly, we noted that the helical parameters of the single-turn helices in the clam-shaped structure fall between those for the helical stem structures, indicating the compatibility of the helical stems and the clam-shaped structure (Supplementary Fig. 8c, d). Thus, we combined both the helical stem and

**Table 1 Cryo-EM data collection and data processing statistics.**

| | NC$_{WT}$ helix-1 (EMDB-30066) (PDB-6M7D) | NC$_{WT}$ helix-2 (EMDB-30065) (PDB-6M7D) | NC$_{cleaved}$ helix (EMDB-30129) (PDB-6M7D) | NC$_{WT}$ Clam-shaped (EMDB-30064) (PDB-6M7D) |
|---|---|---|---|---|
| **Data collection and processing** | | | | |
| Microscope | Titan Krios G$^2$ | Titan Krios G$^2$ | Titan Krios G$^{3i}$ | Titan Krios G$^2$ |
| Voltage (kV) | 300 | 300 | 300 | 300 |
| Camera | Gatan K2 summit | Gatan K2 summit | Gatan K3 BioQuantum | Gatan K2 summit |
| Magnification | 18,000 | 18,000 | 81,000 | 18,000 |
| Electron exposure (e$^-$/Å$^2$) | 40 | 40 | 40 | 40 |
| Defocus range (μm) | 1.5–3 | 1.5–3 | 1.5–3 | 1.5–3 |
| Pixel size (Å) | 0.65 | 0.65 | 0.53 | 0.65 |
| Symmetry imposed | Helical | Helical | Helical | C2 |
| Initial particle images (no.) | 94,916 | 94,916 | 1,686,748 | 436,797 |
| Final particle images (no.) | 4285 | 14,598 | 134,042 | 104,212 |
| Map resolution (Å) | 4.1 | 4.6 | 2.9 | 3.9 |
| FSC threshold | 0.143 | 0.143 | 0.143 | 0.143 |
| Map resolution range (Å) | 4–5.5 | 4–6.5 | 2.5–3 | 3.5–6 |
| **Refinement** | | | | |
| Initial model used (PDB code) | 4UFT | 4UFT | 4UFT | 4UFT |
| Model resolution (Å) | 4.3 | 4.3 | 4.3 | 4.3 |
| FSC threshold | 0.143 | 0.143 | 0.143 | 0.143 |
| Map sharpening B factor (Å$^2$) | −135.99 | −226.73 | −104.14 | −178.62 |
| **Model composition** | | | | |
| Non-hydrogen atoms | 3280 | 3280 | 3280 | 3280 |
| Protein residues | 418 | 418 | 418 | 418 |
| Ligands | 0 | 0 | 0 | 0 |
| **B factors (Å$^2$)** | | | | |
| Protein | 8.5 | 8.5 | 8.5 | 8.5 |
| Ligand | / | / | / | / |
| **R.m.s. deviations** | | | | |
| Bond lengths (Å) | 0.01 | 0.01 | 0.01 | 0.01 |
| Bond angles (°) | 1.025 | 1.025 | 1.025 | 1.025 |
| **Validation** | | | | |
| MolProbity score | 1.78 | 1.78 | 1.78 | 1.78 |
| Clashscore | 15.48 | 15.48 | 15.48 | 15.48 |
| Poor rotamers (%) | 2 | 2 | 2 | 2 |
| **Ramachandran plot** | | | | |
| Favored (%) | 98 | 98 | 98 | 98 |
| Allowed (%) | 2 | 2 | 2 | 2 |
| Disallowed (%) | 0 | 0 | 0 | 0 |

clam-shaped structures together and built an atomic model of double-headed SeV nucleocapsid in which two helices are packed in a tail-to-tail manner (Fig. 1e).

**Assembly mechanism of SeV nucleocapsid.** In double-headed SeV nucleocapsids, the helical stems are stranded by successive protomers together with RNA (Fig. 1c and Supplementary Figs. 2a–c, 4a–c). Similar to other nucleoproteins[6,20–25], SeV nucleocapsids also employ a domain swapping process in which the N-arm and the C-arm interact with neighboring protomers (Fig. 2a, b and Supplementary Fig. 9a, b). The first interface comes from the N-arm from N$_i$, which lies between two α12 helices from N$_i$ and N$_{i-1}$ to assemble into a bundle with three anti-parallel helices. The other side of the α12 helix from N$_i$ is unoccupied and can interact with the N-arm and α12 helix from N$_{i+1}$ (Supplementary Fig. 9a, b). The second interface comprises the C-arm from N$_i$ interacts with α16 helix of N$_{i+1}$ to capture the subsequent nucleoprotein (Supplementary Fig. 9a, c).

Beyond the N-arm/C-arm domain swapping interface, our structural data support the occurrence of a previously unnoticed additional interface between neighboring protomers. Specifically, we observed that the SeV nucleoprotein has an extended loop (Loop$_{20–46}$) connecting the N-arm and the core of NTD (Fig. 1b). Loop$_{20–46}$, along with a loop from CTD (Loop$_{312–320}$) and a loop from the NTD (Loop$_{92–102}$) assembles into a closed-hole adjacent to N-arm (denoted as N-hole) (Fig. 1b and Supplementary Fig. 4d); Loop$_{240–248}$ popping out from NTD of N$_{i-1}$ can become inserted into the N-hole from N$_i$ (Fig. 2a, b). Detailed structural analysis shows that Loop$_{240–248}$ is about 8 Å in diameter, and fits well with the N-hole in size. The surface of the N-hole is overall positively charged, due to the existence of several cationic residues (K$_{21}$, R$_{32}$, K$_{100}$, and K$_{317}$); Loop$_{240–248}$ and its adjacent area have several negatively charged residues including E$_{235}$ and E$_{251}$, which keeps the affinity between them via electrostatic interaction (Fig. 2c). Loop$_{240–248}$ replacement of all negatively charged residues to Alanine abolishes the electrostatic interaction with N-hole, and yields some threading thin filaments (Fig. 2d), hinting at a specific role in the assembly of helical nucleocapsids. Very interestingly, such N-hole-like structures also exist in NDV, PIV5, and MeV[6–9]. Detailed structural analyses indicate the occurrence of electrostatic interaction between N-holes and the extended loops in these paramyxovirus nucleoproteins (Supplementary Fig. 10). Thus, these conserved interfaces between

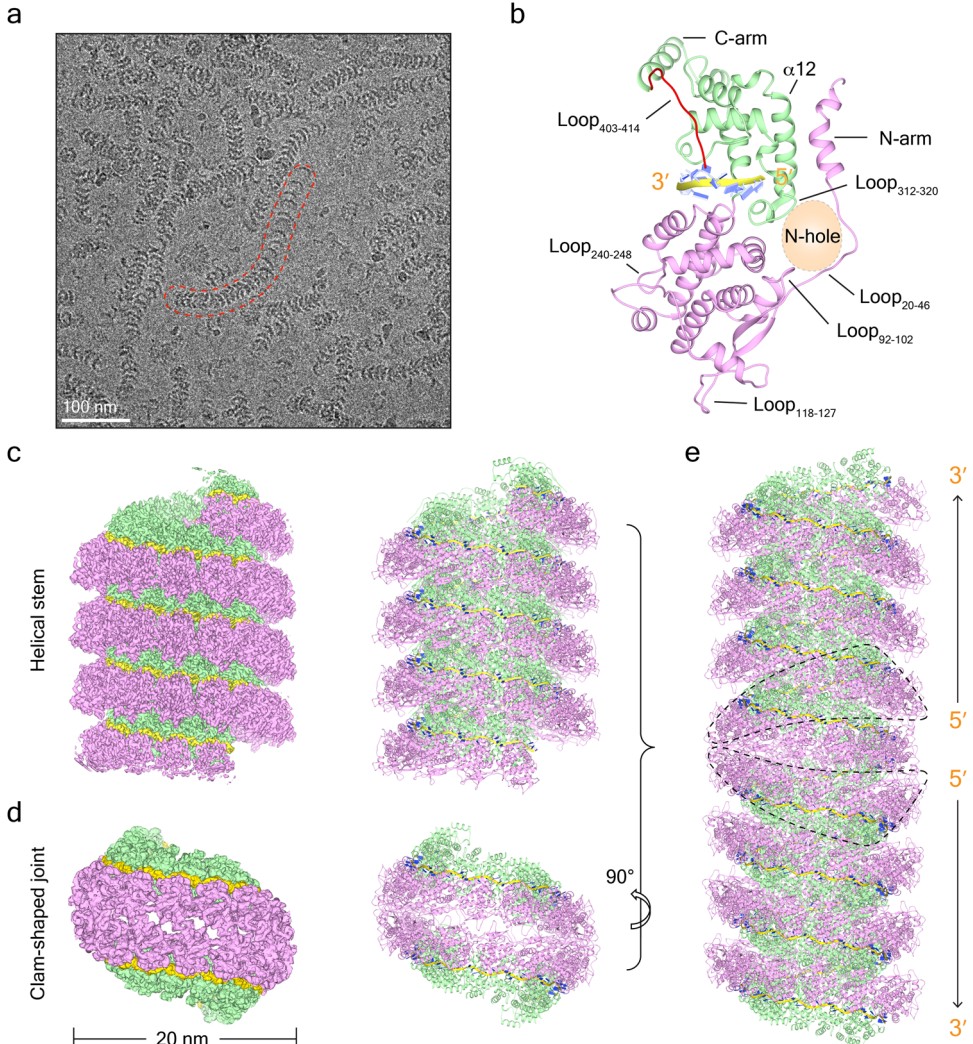

**Fig. 1 Structure of double-headed SeV nucleocapsids. a** A typical cryo-EM micrograph of double-headed SeV nucleocapsids. One curved SeV nucleocapsid with different assembly forms is highlighted in red. **b** The atomic model of one SeV nucleoprotein with the accompanied RNA. NTD and CTD are colored in pink and green, respectively. Loop$_{403-414}$ in N-tail is colored in red. N-hole formed by Loop$_{20-46}$, Loop$_{92-102}$, and Loop$_{312-320}$ is depicted in light orange. RNA is depicted with the backbones in gold and the bases in blue. The same color code is used for the rest of the figures. **c** 3D reconstruction of the helical stem of double-headed SeV nucleocapsids and the respective atomic model. **d** 3D reconstruction of the clam-shaped joint of double-headed SeV nucleocapsids and the respective atomic model. **e** Pseudo-atomic model of the double-headed SeV nucleocapsid. The clam-shaped structure is depicted in a dashed line and the RNA direction of two embedded RNA strands is labeled.

N-holes and the extended loops resemble a gate latch and bolt, and apparently function to tightly anchor the positions of neighboring nucleoprotein protomers. Therefore, N-hole adopts the same domain swapping process as N-arm and C-arm, and contributes to the assembly of helical nucleocapsids in the family of *Paramyxoviridae* (Fig. 2e).

It is notable that all of these interfaces involved by N-arm, C-arm, and N-hole occur in the interior of SeV nucleocapsids. Similar to most nucleocapsids[6,7,20–26], RNA cleft between NTD and CTD is facing the outsides of SeV nucleocapsids (Supplementary Fig. 9b, d). Recombinant SeV nucleoproteins can enwrap RNA from host cells, which is supported by our data of the absorbance of OD260/OD280 at ~1.2 during nucleoprotein purification (Supplementary Fig. 1a) and clear EM densities of RNA in both high-resolution structures of NC$_{WT}$ and NC$_{cleaved}$ (Fig. 1c and Supplementary Fig. 4c). In SeV nucleocapsids, negatively charged RNA interacts with nucleoprotein residues K$_{180}$, R$_{195}$, and R$_{354}$, and six nucleotides are precisely associated with each protomer, with a typical "3-base-in" and "3-base-out"

conformation (Supplementary Fig. 9d and Supplementary Movie 3).

**N-tail correlates with SeV nucleocapsid curvature**. Positioned immediately following the C-arm, the N-tail of Hendra virus nucleoprotein is known to function in the regulation of gene replication and transcription via binding to the C-terminal X domain of phosphoprotein[27]. To date, no structural information is available about the N-tails of the paramyxovirus nucleoproteins, likely owing to the intrinsic flexibility of these tails[28]. In our high-resolution structures of SeV NC$_{cleaved}$, the first 12 residues (403–414) of the nucleoprotein N-tail are resolved (Figs. 1b and 3a). These 12 residues assemble into a loop (Loop$_{403-414}$) that is orientated in the direction of the C-arm, extends about 36 Å, and points toward the outsides of nucleocapsids (Fig. 3b). Considering that the inner contact sites between neighboring rungs in helical stems of SeV nucleocapsids are filled with C-arm and Loop$_{403-414}$, there is virtually no free space available to

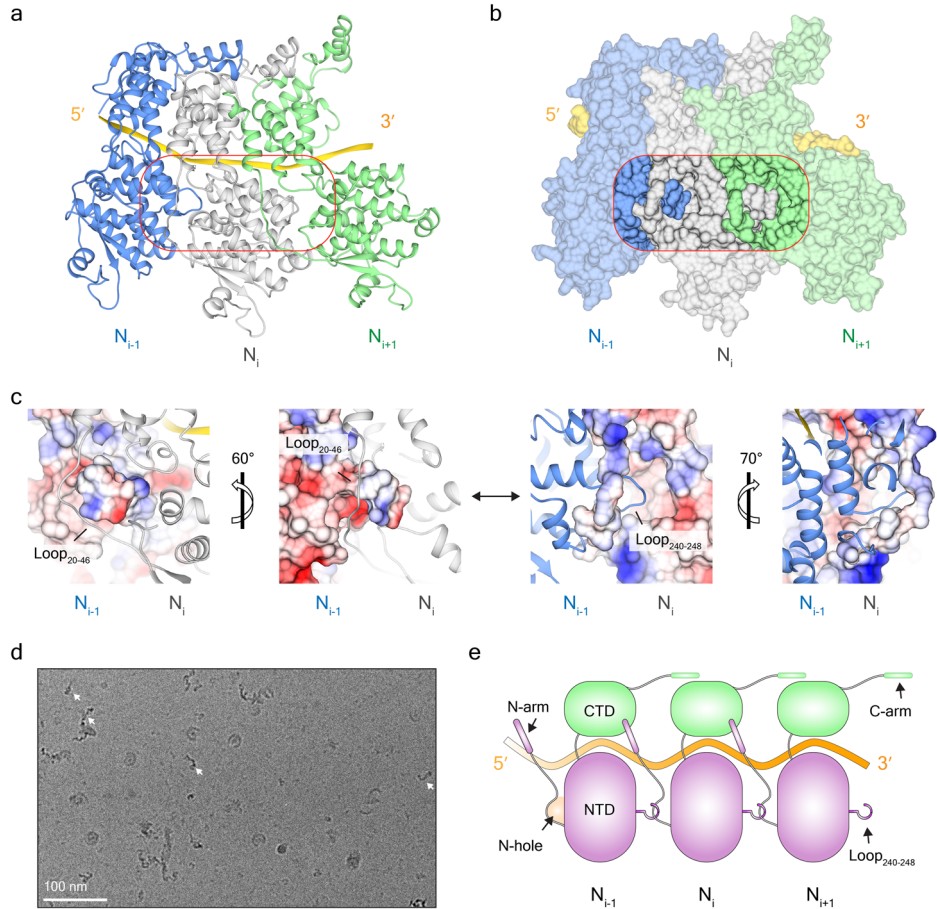

**Fig. 2 The assembly mechanism of the helical stem of double-headed SeV nucleocapsids. a** The atomic model of three neighboring protomers in the helical stem. Protomers are colored in blue, gray, and green, respectively, and RNA is colored in gold. An unnoticed swapped interface between neighboring protomers is boxed in red. **b** Molecular surfaces of three neighboring protomers with the same view as in **a**. The same box is applied for the new interface. The transparency values of the boxed area and the other part are set to 0% and 40%, respectively. **c** Electrostatic interaction between N-hole from $N_{i-1}$ and $Loop_{240-248}$ from $N_i$. On the left two images, $N_{i-1}$ is represented in the contour surfaces of electrostatic potential, and $N_i$ is displayed in the ribbon. On the right two images, $N_{i-1}$ is displayed in the ribbon, and $N_i$ is represented in the contour surfaces of electrostatic potential. **d** Threading thin filaments formed by $Loop_{240-248}$ replacement mutant. Typical threading filaments were marked with arrows. **e** A model illustrates domain swapping mechanism in neighboring protomers.

accommodate the residual N-tail (415–524) in a way that would allow it to turn back and penetrate the contact regions to the inside of nucleocapsids (Fig. 3a, b).

The cavity between two neighboring rungs of SeV nucleocapsids has a volume of ~11,000 Å³, which is too small to accommodate the residual N-tail of ~18,000 Å³. Some regions of the N-tail such as the MoRE motif, are expected to exist at the outsides of nucleocapsids and bind phosphoprotein to regulate gene replication and transcription[3,29]. Purified SeV $NC_{WT}$ are frequently curved (Fig. 3c), no matter long or short, just as reported in MeV nucleocapsid[30]. The persistence length for SeV $NC_{WT}$ has a value of ~288 nm. Interestingly, the removal of most N-tail from SeV $NC_{WT}$ either after 5 weeks storage at 4 °C or via trypsin digestion yields straighter filaments as revealed by cryo-EM (Fig. 3d and Supplementary Fig. 11a, b), with a bigger persistence length value at ~877 nm. Semi-quantification analysis on trypsin digested SeV nucleocapsids at different time points further shows that the straightening of nucleocapsid is tightly correlated with the removal of N-tail (Fig. 3e and Supplementary Fig. 11c).

**Hyper-closed SeV clam-shaped structure.** Double-headed SeV nucleocapsid assembles in a tail-to-tail manner, similar to our

previous observation for the clam-shaped NDV nucleocapsid structure[9]. However, compared to the relatively loose interface in NDV nucleocapsids, the clam-shaped joint in SeV nucleocapsids adopts a tightly crisscrossed pattern; this tighter and engaged pattern clearly impacts the capacity for lateral sliding between two opposite single-turn spirals (Fig. 4a). The distance between two opposite protomers in the SeV clam-shaped structure is about 30 Å, only half of the distance compared to the NDV clam-shaped structure[9]. Accordingly, the empty space between two opposite protomers is reduced from ~1045 Å² in NDV to only ~330 Å² in SeV (Fig. 4a and Supplementary Movie 4). Given the much smaller space between two single-turn spirals, SeV clam-shaped structure nucleocapsid is depicted as a "hyper-closed" form.

The contact site between two opposite protomers of SeV clam-shaped structure comprises residues from 118 to 127 ($Loop_{118-127}$) (Fig. 4b). The top of $Loop_{118-124}$ from $N_i'$ is negatively charged, which binds to positively charged areas composed of $R_{32}$, $K_{100}$, and $R_{129}$ from the opposite protomer $N_i$ via electrostatic interaction. There is another interface involved by several residues including positively charged $K_{108}$, $K_{111}$, and $K_{121}$ from $N_i'$, and negatively charged $Loop_{118-127}$ from the opposite protomer $N_{i-1}$ (Fig. 4b and Supplementary Movie 5). Due to the two-fold symmetry owned by the clam-shaped structure, such

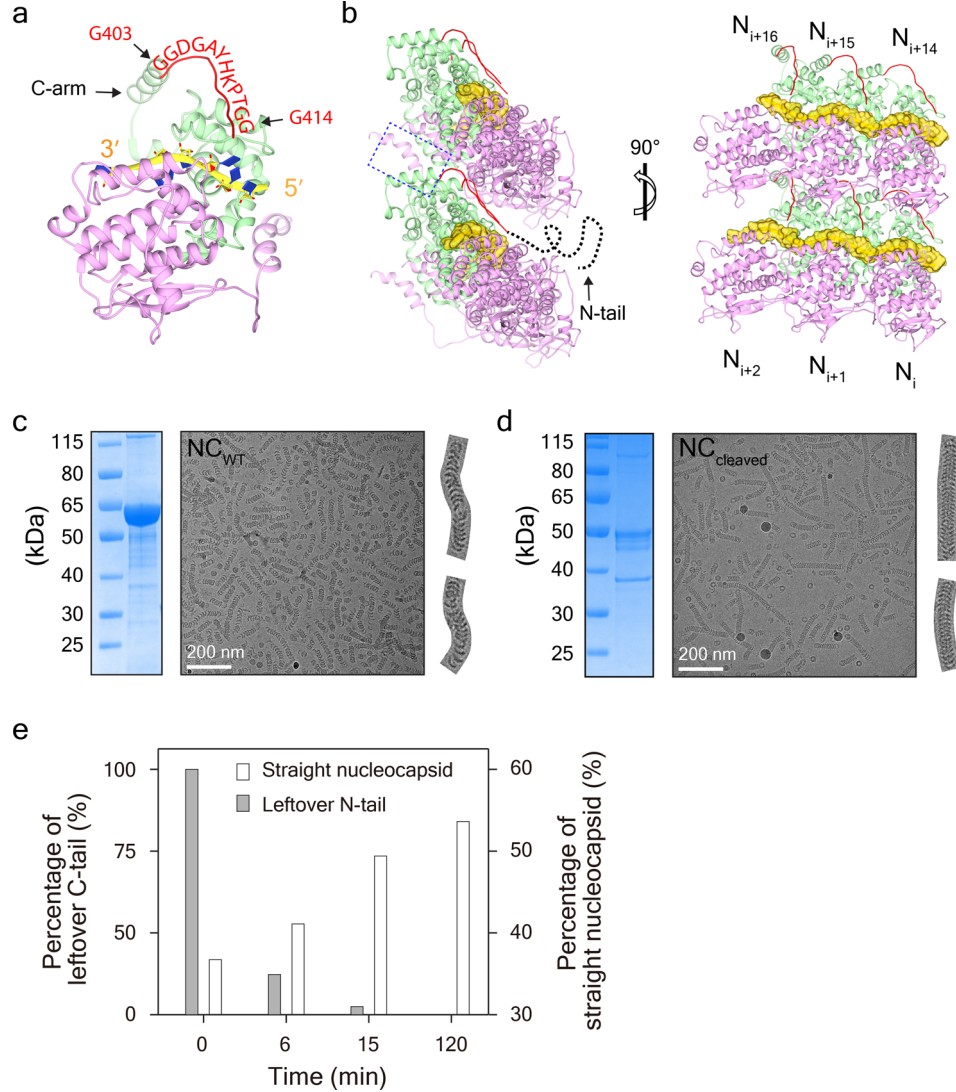

**Fig. 3 N-tail correlates with nucleocapsid curvature. a** The first 12 residues (403–414) of N-tail are resolved in SeV NC$_{cleaved}$. Loop$_{403–414}$ is colored in red and the residue names are labeled along the loop. **b** Different views of 6 protomers from two neighboring rungs are shown. The inner contact site between neighboring rungs is marked in a dashed blue box. The unidentified residues (415–525) are denoted as dotted curved lines. **c** Curved nucleocapsid formed by NC$_{WT}$. Two typical curved filaments are cut and zoomed in. **d** Straight nucleocapsid assembled by NC$_{cleaved}$. Two typical straight filaments are cut and zoomed in. **e** The straightening of nucleocapsid is tightly correlated to the removal of N-tail from SeV nucleocapsid.

interface and the interaction force will be doubled to enhance the clam assembly.

Loop$_{118–127}$ seems only involved in the maintenance of clam-shaped joints between two helical stems. In the helical stems, Loop$_{118–127}$ does not have significant interaction with neighboring protomers from upper/lower rungs (Fig. 4c). Residues between 118 and 127 are replaced with all Alanine and the loop mutant was purified as usual. Loop mutation abolishes the formation of double-headed filaments but keeps single-headed filaments (Supplementary Fig. 12). Mutation mapping on individual residue between 118 and 127 shows that F$_{118}$ has a marked influence on clam-shaped structure assembly, which might be caused by a conformational change of Loop$_{118–127}$ and the derived charge distribution change (Supplementary Fig. 12). Detailed sequence alignment shows a hydrophobic residue "F/P/ M" followed by a positive-charged residue "R/K" in Loop$_{118–127}$, highly conserved in several viruses including SeV, NDV, PIV5, and NiV, which indicates that clam-shaped structures might be popular in members of paramyxoviruses (Fig. 4d and Supplementary Fig. 13).

## Discussion

In the present study, we detected a double-headed SeV nucleocapsid purified from *E. coli*, and resolved both its helical stems and its clam joint at near-atomic resolutions. Similar to the clam-shaped structure from NDV[9], SeV utilizes its clam-shaped structure as the nucleator for the formation of double-headed nucleocapsids, wherein paramyxovirus nucleoproteins enwrap viral RNA to form a highly stable assembly. Not limited to our discoveries from NDV and SeV, the formation of clam-shaped assemblies of nucleoproteins was recently reported in NiV after overexpression in bacteria, which is also mediated by inter-subunit interactions involving several nucleoprotein loop regions[19]. Considering that not all viruses in the family of *Paramyxoviridae* have been found to contain such structures, the formation of clam-shaped structures might be due to preparative techniques. To check this, SeV nucleoprotein was over-expressed in HEK293F cells. Following a two-step ultracentrifugation process of cell lysate, fractions containing nucleoproteins were directly subjected to cryo-EM analysis. Notably, double-headed nucleocapsids (with the exact same diameter at 19 nm) were also

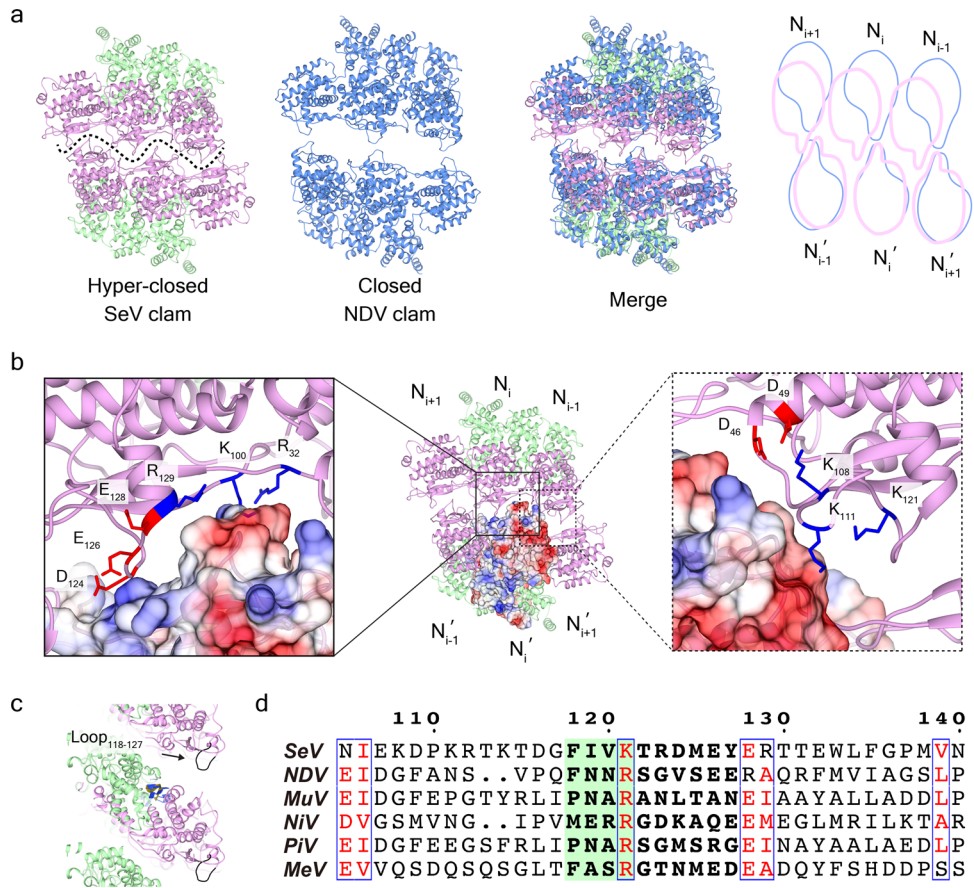

**Fig. 4 Hyper-closed clam-shaped joint in double-headed SeV nucleocapsid. a** Hyper-closed SeV clam-shaped assembly compared with NDV clam-shaped structure. The interface between opposite protomers in SeV clam-shaped structure is depicted in the dashed curve line. **b** Interface analysis of SeV clam-shaped structure. Two contact sites are marked in the solid box (left) and dotted box (right), respectively. Positively charged residues in the upper helix are labeled and the contour surface of electrostatic potential is shown in the opposite rung. **c** Loop$_{118-127}$ is not involved in the assembly of the helical stem. **d** Sequence alignment of Loop$_{118-127}$ of SeV nucleoprotein in members of paramyxoviruses. The critical residues from 118 to 121 are shaded in green.

present among nucleoproteins isolated from HEK293F cells (Supplementary Fig. 14a).

Furthermore, we attempted to verify the occurrence of clam-shaped structures in SeV virion. After Triton lysis of SeV virions, SeV nucleocapsids spread out on cryo-EM grids and clam-shaped structures surrounded by two herringbone structures packing in a tail-to-tail mode can be captured (Supplementary Fig. 14b). Electron tomography on intact SeV virions and tomography averaging of nucleocapsids are supposed as a better way to verify the occurrence of clam-shaped structures in situ.

Although small quantities of double-headed nucleocapsid structures have been visualized in multiple paramyxoviruses, their biological relevance is unclear at this point. Such structures are speculated to confer benefits for genome stability, polyploid genome organization or genome condensation[14,31–35]. Thus, it will be interesting to explore both the occurrence of such double-headed structures throughout the family of *Paramyxoviridae* and to examine the specific cellular conditions in which this nucleocapsid type is preferentially formed.

There is extensive in vitro and in vivo evidence showing that curved nucleocapsids occur commonly in paramyxoviruses, with examples of curvature for both single-headed and double-headed nucleocapsids[6,7,17,29,32,36,37]. In one SeV nucleocapsid, clam-shaped joint and straight/condensed filaments coexist with loosed or even uncoiled filaments (Fig. 5a). Our results establish that partial removal of the N-tail yields much straighter filaments, and

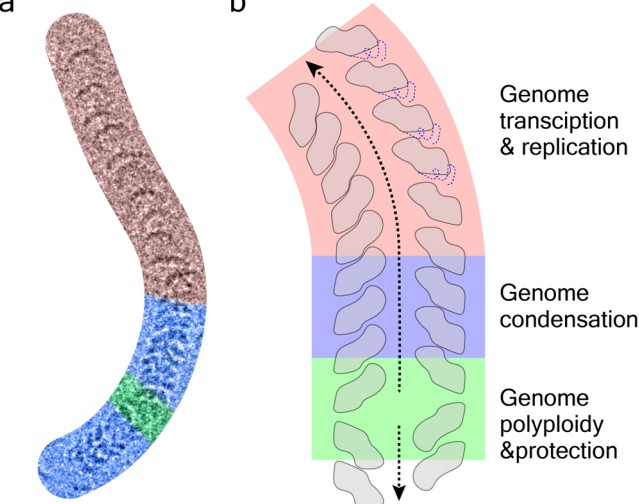

**Fig. 5 A model illustrating the hypothesized distinct functions of different assembly forms. a** Clam-shaped structure (light green), helical stems (light blue), and loosed coiled filament (light red) in one SeV nucleocapsid are highlighted. **b** A model illustrating the proposed distinct functions of different assembly forms. The same color strategy is used as in **a**.

demonstrate that N-tail is involved in curvature regulation of double-headed nucleocapsids. Comparing to straight fragments, curved ones might have imbalanced states of N-tail between the inside side and the opposite outside side[38]. This difference might get the phosphoprotein binding site exposed, and thereafter regulate gene transcription and replication. Actually, highly curved or even bent nucleocapsids will be vital for genome packing into virions. All these results point to a bold hypothesis that different assembly forms in double-headed SeV nucleocapsid represent distinct functions, which might be applied to other paramyxoviruses (Fig. 5b).

## Methods

**Sequence alignment**. Nucleoprotein sequences including Sendai virus (NP_056871.1; access numbers were obtained from the NCBI protein database), Newcastle disease virus (YP_009513194.1), mumps virus (NP_054707), Nipah virus (NP_112021.1), Parainfluenza virus 5 (YP_138511.1), and measles virus (NP_056918.1) were downloaded from PubMed in FASTA format. Geneious was used to align the sequences[39], and the alignment was displayed via ESPript[40]. The secondary structure prediction on N-tail of SeV nucleoprotein was fulfilled in PSIPRED[41].

**Plasmids and gene expression**. $N_{WT}$ was cloned into pCAGGS plasmid with 6× His-tag on C-termini for protein expression in mammalian cells. $N_{WT}$ and its derivatives with two 6× His-tags on both N- and C-termini were synthesized into pET28b plasmids for gene expression in *Escherichia coli*. All plasmids were verified via gene sequencing before gene expression.

$N_{WT}$ in pCAGGS plasmid was transfected into HEK293F cells with Lipofectamine 2000, when the cell density reached $4 \times 10^6$ cells/mL. The transfected cells were cultured at 37 °C, 6% $CO_2$, 180 rpm for another 3 days and then harvested via centrifugation at $1000 \times g$ for 15 min.

$N_{WT}$ and its derived mutants expressed in *E. coli* BL21 (DE3) cells were used for high-yield protein expression. In details, cells containing the respective plasmids were cultured in LB media at 37 °C until OD600 reached 0.8. Target proteins were induced with 1 mM IPTG (isopropyl-β-D-1-thiogalactopyranoside) at 16 °C, 220 rpm for 20 h. The cells were harvested by centrifugation at $4680 \times g$ for 20 min to obtain the cell pellets.

**Protein isolation and purification**. $N_{WT}$ in HEK293F cells were lysed in lysis buffer (20 mM Tris-HCl (pH7.4), 150 mM NaCl, 1 mM $CaCl_2$, 5 mM β-Mercaptoethanol and protease inhibitor cocktail (Roche, USA)). The lysate was clarified by centrifugation at $3200 \times g$ for 20 min. The supernatant was loaded onto a double sucrose cushion (90% (w/v) sucrose is beneath 20% (w/v) sucrose) and ultracentrifuged in a Beckman SW32 Ti rotor at $130,000 \times g$ for 3 h. 1 mL fractions were manually collected via bottom puncture, 10 μL of which were subjected to western-blot analysis. Fractions containing nucleoproteins were collected into dialysis tubing with the molecular weight cutoff at 1 MDa, and dialyzed overnight in the sucrose-free lysis buffer to remove sucrose. The dialyzed sample was concentrated and subjected to another round of ultracentrifugation. Specifically, 1.5 mL of the condensed protein was loaded onto a 25–60% continuous sucrose gradient and was spun in a SW41 Ti rotor at $160,000 \times g$ for 4 h. Samples were automatically fractionated by Fraction FC-203B collector (Gilson, USA), and the fractions containing nucleoproteins were collected based on western-blotting against 6×His-tag before overnight dialysis to remove sucrose. The dialyzed sample was concentrated and immediately applied to cryo-EM sample preparation.

$N_{WT}$ and its derived mutants expressed in *E. coli* were purified via tandem affinity and gel-filtration chromatography. Specifically, pelleted cells were resuspended in simplified lysis buffer (20 mM Tris-HCl (pH 7.4), 150 mM NaCl), and disrupted via ultrasonic homogenizers (JNBIO, China). After centrifugation at $47,850 \times g$ for 30 min, the supernatant was loaded onto a 5 mL HisTrap™ HP column (GE Healthcare LifeSciences, USA), preequilibrated with simplified lysis buffer. The column was washed with 50 mL simplified lysis buffer with a step gradient of imidazole at 20, 50, and 100 mM. Finally, proteins were eluted using a simplified lysis buffer containing 500 mM imidazole. Proteins were concentrated and loaded onto a 24 mL Superose 6 increase 10/300 GL chromatography column (GE Healthcare Lifesciences, USA) preequilibrated with simplified lysis buffer. 0.2 mL fractions were collected, 10 μL of which were subjected to SDS-PAGE analysis.

All protein samples were freshly made in the following assays except that purified $N_{WT}$ was stored at 4 °C for 5 weeks for the cleavage assay.

**Negative stain EM**. 4 μL of protein samples (~0.1 mg/mL) were applied to glow-discharged EM grids covered with a thin layer of continuous carbon film and stained with 2% (w/v) uranyl acetate. Negatively stained grids were imaged on a Talos L120C TEM (Thermo Fisher Scientific, USA) operating at 120 kV. Images were recorded at a magnification of ×73,000 and a defocus set to about −2 μm, using a Ceta™ 16M camera (Thermo Fisher Scientific, USA).

**Cryo-EM data collection**. 3.5 μL of samples (~1 mg/mL) were applied to glow-discharged holey grids R2/1 (Quantifoil, Ted Pella) with a thin layer of the continuous carbon film. The grids were blotted using a Vitrobot Mark IV (Thermo-Fisher Scientific, USA) with a 1 s blotting time, force level of 2, and the humidity of 100% at 16 °C, immediately plunged into liquid ethane and stored under liquid nitrogen temperature for future cryo-EM imaging. Cryo-EM grids were examined in the low-dose mode on a Talos L120C TEM for screening or instant imaging. Snapshots were taken at a magnification of ×73,000 and a defocus set to about −2 μm, using a Ceta™ 16 M camera.

Data collections on good grids were performed on two Titan Krios microscopes: Titan Krios $G^2$ TEM (ThermoFisher Scientific, USA), equipped with a K2 Summit direct electron detector (Gatan, USA), which was used in the super-resolution mode with a pixel size of 0.65 Å; Titan Krios $G^{3i}$ TEM (ThermoFisher Scientific, USA), equipped with a K3 BioQuantum direct electron detector (Gatan, USA), which was used in the super-resolution mode with a pixel size of 0.53 Å. Special care was taken to perform a coma-free alignment on the microscopes and detailed data collection conditions were listed in Table 1. All the images were collected under the SerialEM automated data collection software package[42], and data sets from two Titan Krios scopes were subjected to data analysis, separately.

**Cryo-EM data processing and 3D reconstruction**. Before image processing, raw frames were aligned and summed with dose weighting under MotionCor2.1[43] and the CTF parameters were determined by CTFFIND-4[44]. Image processing was mainly performed in RELION 3.1[45], and different reconstruction strategies including helical reconstruction and single-particle analysis were applied to helical stems and clam-shaped assemblies of double-headed SeV nucleocapsid, respectively. The detailed workflows for helical reconstruction and single-particle analysis after two-dimensional classification are shown in Supplementary Fig. 3 and Supplementary Fig. 6, respectively.

1. Single-particle analysis
   Ring-like particles were picked automatically in RELION and joints between two helices in each filament were manually picked in addition to increasing side views of clam-shaped structures. Obvious junks were excluded by reference-free 2D classification. NDV clam-shaped structure (EMDB, EMD-9793)[9] and half of the clam (one single-turn spiral) were low-pass filtered to 60 Å and chosen as the references for 3D multiple reference alignments. Only classes with the whole clam-shaped structures were selected for further 3D auto-refinement. A 3D map was obtained after 3D refinement with the enforced two-fold symmetry, filtered, and sharpened with RELION post-processing session. An overall resolution was estimated at 5.6 Å based on gold standard Fourier Shell Correlation (FSC) 0.143 criteria.
   Local resolution of the 3D structure was measured with RELION, and protomers further away from the gap of the clam-shaped structure are much better resolved than those closer to the gap due to the structural wobbling. To keep enough signal for accurate alignment, 5 pairs of consecutive protomers from clam-shaped structures, which are furthest from the gap, were chosen for local refinement. To make full use of the local circular symmetry, the particle center was switched to each pair of protomer via turning the center one protomer further using our own script. Thus, the total particle number was increased five times with the distributed Euler angles. The expanded particle set was subjected to 3D refinement with a local angular search for more accurate alignment and another round of 3D classification without alignment was applied to reduce the heterogeneity. Particles from classes with the best resolution were subjected to the 3D refinement focusing on the five pairs of protomers with a local angular search of ±6°. The final reconstruction was determined with the resolution of 3.9 Å by gold standard FSC 0.143 after polishing.
2. Helical reconstruction
   Start and end points of helical stems of double-headed SeV nucleocapsids are manually specified and particles were extracted every 7 asymmetric units (about 90% overlap) along the helices. Junks and curved fragments were removed based on 2D classification. The initial model was synthesized from one single-turn spiral of SeV clam-shaped structure, which is low-pass filtered to 30 Å for 3D classification. Helical symmetry was applied during 3D classification and classes were merged into one or more groups, depending on their helical rise, twist and resolution. For $NC_{WT}$, particles belonging to different classes were subjected to refinement respectively. For $NC_{cleaved}$, since the variation of rise and twist was within a relatively narrow range and the particle distribution among classes was unstable through iterations, only the classes with more promising resolution were combined and subjected to refinement and Bayesian polishing. The final reconstructions were filtered and sharpened in RELION post-processing session. The resolutions were determined by gold standard FSC 0.143. Detailed helical twist and rise for each dataset are listed in Supplementary Fig. 3.

**Model building and structural analysis**. The homology model of SeV nucleoprotein and RNA was generated by Modeller[46] using the crystal structure of measles virus nucleoprotein (RCSB, PDB-4UFT) as the template[6]. Pseudo-atomic model of SeV nucleoprotein was flexibly docked into the EM density of $NC_{cleaved}$ at

the 2.9 Å resolution using Rosetta[47] and Flex-EM software[48]. Residues (403–414) of N-tail were manually traced and placed against densities of $NC_{cleaved}$ and trypsin cleaved nucleocapsid with better resolved 403–408 with Coot[49]. The atomic model including both nucleoprotein and RNA was further optimized for better local density fitting using real-space refinement in Phenix[50]. The final SeV $NC_{cleaved}$ atomic model was duplicated to build the atomic model of SeV $NC_{cleaved}$ helical stem. This atomic model was also docked as a rigid body to helical stems and clam-shaped structures of double-headed SeV nucleocapsids using the University of California, San Francisco Chimera package[51].

The extra EM densities enwrapped between NTD and CTD in all reconstructions were assigned as RNA and docked using poly-Uracil in Coot due to the unspecific binding of nucleoprotein to RNA[49].

The structural analysis including surface electrostatic distribution and structural superimposition was fulfilled in UCSF Chimera[51].

**Trypsin enzymatic assay**. Both $N_{WT}$ and $N_{F118A}$ were treated with trypsin to test the susceptibility. A 40 μL mixture of $N_{WT}$ or $N_{F118A}$ (1 mg/mL) with trypsin (0.003 mg/mL) was incubated at 4 °C and 5 μL fraction was taken out for SDS-PAGE analysis at 0, 3, 6, 10, 15, 120, and 240 min, respectively. Quantitation of bands at ~65, 50, and 35 kDa were analyzed densitometrically by CLIQS (TotalLab, UK).

For trypsin digestion assay on $N_{WT}$ under the same condition, another 3 μL fractions were taken out for cryo-EM analysis at 0, 6, 15, and 120 min, and 25 images were captured at the magnification of ×57,000 for each sample. Numbers of straight nucleocapsids were counted, and the percentage of straight nucleocapsids among all filaments was calculated at different digestion time points.

**Nucleocapsids isolated from SeV virion**. Wild-type SeV virion was propagated in embryonated 9-day-old chicken eggs, as described[52]. SeV infected allantoic fluid was harvested and stored under −80 °C until use.

The thawed allantoic fluid was centrifuged for 15 min at 4000×g under 4 °C to remove the crude debris. The supernatant was subjected to another round of centrifugation for 4 h at 48,000×g under 4 °C, and the resultant pellet was resuspended with PBS buffer (50 mM $NaH_2PO_4$, 50 mM NaCl, pH 7.2) and then loaded onto the 25–65% (w/v) continuous sucrose gradient column. After the 18 h ultracentrifugation at 150,000 g under 4 °C, the visible layer was collected and dialyzed overnight in PBS buffer to remove sucrose. The virion suspension was concentrated and lysed with 2% (v/v) Triton X-100 for 6 h under 4 °C, 3.5 μL of which was immediately applied to cryo-EM sample preparation. 500 good micrographs were collected on Titan Krios $G^{3i}$ TEM (ThermoFisher Scientific, USA), equipped with a K3 BioQuantum direct electron detector (Gatan, USA),

**Persistence length analysis**. Persistence length analysis on SeV $NC_{WT}$ and $NC_{cleaved}$ was performed in ImageJ as previously reported[53]. Briefly, 200 filaments per each were traced using an ImageJ plugin, JFilament[54], and persistence length analysis on SeV $NC_{WT}$ and $NC_{cleaved}$ were calculated in ImageJ.

**Reporting summary**. Further information on research design is available in the Nature Research Reporting Summary linked to this article.

## Data availability

The cryo-EM density maps of double-headed SeV nucleocapsids were deposited in Electron Microscopy Data Bank (EMDB) with the accession numbers 30064 (clam-shaped structure), 30065 (helical stem-1), 30066 (helical stem-2), and 30129 ($NC_{cleaved}$), respectively. And the atom coordinates of a single N subunit were deposited in the Protein Data Bank (PDB) with the PDB ID code 6M7D. All other data are available in the main text or the supplementary materials or available from the corresponding author upon reasonable request.

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

## Acknowledgements

We thank Prof. Yifan Cheng from UCSF and Prof. Yun Bai from ShanghaiTech University for thorough discussions. We are grateful to Kang Li, Dianli Zhao, and Ceng Gao from the CryoEM facility for Marine Biology at QNLM for our cryo-EM data collection. We also thank the Electron Microscopy Facility of ShanghaiTech University for sample preparation and data collection. This work was supported by the National Key R&D Program of China, 2017YFA0504800 (Q.S.), National Key R&D Program of China, 2018YFC1406700 (Q.S.), National Natural Science Foundation of China, 31870743 (Q.S.) and Young Scientists Fund of the National Natural Science Foundation of China, 31800617 (R.W.).

## Author contributions

N.Z. expressed and purified proteins, and prepared and screened EM grids. N.Z., H.S., and M.L. collected cryo-EM data sets. H.S. and N.Z. did data processing and carried out model building and refinement with the assistance of Y.L. T.L., L.Y., X.C, R.L., X.S., L.Q., R.W., and W.S. helped with input and discussion during the course of the work. N.Z., H.S., and Q.S. analyzed all the data. Q.S. supervised the work. Q.S. wrote the manuscript with input from N.Z., H.S., and T.L.

## Competing interests

The authors declare no competing interests.
