## [Peer Review File · Communications Biology]

Reviewers' comments:

Reviewer #1 (Remarks to the Author):

In the presented manuscript, Zhang et al present the reconstruction of the Sendai Virus N-RNA complex at atomic resolution, including the helical part and a clam structure previously described by the same authors for NDV. Also, they demonstrate the presence of an additional, ring like structural motive stabilising N-protein interactions. In general, the manuscript is well written and the figures are clear and easy to understand. Also, the manuscript is accompanied by a substantial amount of supplementary information and movies to better describe the findings for the interested reader.

However, in the opinion of this reviewer, some points should be addressed before the manuscript can be considered for publication.

A) In the introduction, please mention that SeV is part of the Paramyxoviridae and indicate which structures have already been solved of the N-RNA complex in this virus family and how different they are at the aa versus the structural level.

B) In the section line 64-72, you talk about the complex purified from HEK cells. However, as far as I understand, all the data for further structural and functional analysis, were generated employing protein expressed in bacteria. I find this confusing, as it makes it more difficult for the reader to follow your line of thought and immediately raises the question why you didn't work with the protein expressed in eukaryotic cells. So please either leave the data on the HEK cells out or explain / comment / discuss the approach you chose.

C) line 73-77 are difficult to understand. Please rephrase

D) line 81-84: please explain how cleavage was achieved (trypsin?, which is mentioned later in the manuscript) and where approximately cleavage took place. If C-tail refers to specific aa, please indicate them here.

E) line 94ff: the reconstruction of the clam structure to my understanding was performed on ring like structures. Please explain in the text how you validate that the structure observed between helical stems is indeed the same as the rings?

F) line 100ff: As I am not sure I understood correctly, so the reconstruction strategy was to use 5 protomers (please indicate what protomers further away from the gap refers to, especially indicate what the gap is) and then use the circular symmetry and turn them one monomer further? Please explain in more detail so that it is easier for the reader to understand.

G) In the paragraph starting line 112, please include some structural comparison to already resolved nucleoprotein structures from for example Measles and NDV to highlight similarities and differences.

H) line 174: To my understanding, this analysis is a quantification, but not a statistical analysis.

I) line 197ff: Was this analysis performed in material purified from HEK cells or bacteria?

J) line 205ff: The mentioned FXXR/K motive is only conserved in SeV, NDV and MeV in the aligned sequences. Otherwise, it is a R/K motive, at least to my understanding. Please explain further or correct this statement.

K) line 225ff: Please mention in which members of the Mononegavirales you found these clamp structures and in which context (protein purified from over expression / purified from viruses or virus infected cells / visualised directly in the virus particle)

L) Discussion: The finding of double headed nucleocapsids for viruses not encoding a segmented genome is intriguing. Could you please further elaborate on the possible function, what your underlying evidence is to propose the model shown in Fig 5, whether these structures are present in virus particles and virus infected cells and how the occurrence of these structures could be affected by the production system / purification procedure?

Reviewer #2 (Remarks to the Author):

This manuscript reports the structural analyses of the nucleoprotein (NP) from Sendai virus (SeV), a negative strand RNA virus (NSV), by cryo EM image reconstruction. The coding sequence for the NP has been used to construct an expression vector to express a recombinant protein in HEK293F cells or E. coli. Purification by ultracentrifugation produced assemblies of the NP in complex with cellular RNA, which may have structural features representing the viral nucleocapsid. By application of cryo EM image reconstruction techniques, several structures were determined at different resolutions (However, it is not clear which structure is derived from which protein assembly. The authors randomly use the same term to describe different structures). Primarily, two structures are reported: a helical assembly, and a complex of two spiral rings.

The reported structure model was built and refined starting from the coordinates of measles virus (MeV) nucleoprotein, a closely homologous structure. By visual inspection, the reported structure of the SeV helical complex of NP-RNA is nearly identical to that of MeV. Without direct comparisons between the two structures, it is not clear what new features are discovered in SeV nucleocapsid. However, there are too many incorrect statements in the manuscript that are inconsistent with published data and the established nomenclature/knowledge in the field. Without careful revision and thorough literature citation, this manuscript does not have the quality to be published in a respectable scientific journal. The authors are advised to read published literature thoroughly to have a good understanding of the field and the subject in discussion. Without such knowledge, the presentation by the authors is scientifically weak, or even useless.

(1) "tail". Sendai virus is a prototype NSV that has been studied for many decades in the field. A set of nomenclature has been well established to describe the structure and function of SeV NP. The domain at the C-terminus of SeV NP has been named N-tail (residues 402-524). It is completely foolish for authors to rename it C-tail. In the section named "C-tail regulates nucleocapsid curvature" by the authors, erroneous statements are numerous.

e.g. "the C-tail of mononegaviruses is known to function in the regulation of gene replication and transcription via binding to phosphoprotein^{3,23-26}." The works cited in this sentence did not use a nucleocapsid protein that has a N-tail. In addition, not all viruses in Mononegavirales have a NP that contains a N-tail.

e.g. "These 12 residues assemble into a loop (Loop403-414)". It is not understandable why the authors named this strand "a loop". More critically, the conformation of this strand in a N-tail removed protein may be totally different from that in the full length protein. "our data provide direct evidence that the "C"-tail extends outside of the nucleocapsids (Fig. 3a, b)". However, published data showed that the N-tail of the full length NP may be inside or outside the helical nucleocapsid. "Loop403-414 to extend about 36 Å from the C-arm, where it reaches the RNA cleft between the NTD and CTD". The grammar is incorrect. This reaching to the RNA cleft would not occur when the N-tail is intact. "The undulating surfaces of CTDs positioned between two neighboring rungs of SeV nucleocapsids force Loop403-414 to extend about 36 Å from the C-arm". How does a surface "force" a peptide to extend? Where is the energy from? "Loop403-414 functions like a lid to cover RNA cleft, which will offer thorough protection against possible damage to RNA". How can a strand be a lid? According to Fig. 3a, the position of residues 403-414 is far away from the packaged RNA. "damage to RNA"? What damage? Radiation?

e.g. "which directly indicates that C-tail regulates nucleocapsid curvature." The curvature of the NSV nucleocapsid is completely dependent on environment, including interactions with other biological molecules, ionic strength, or pH. The artificial curvature observed by the authors is dependent on the presence of N-tail under their study conditions. However, this has no biological

relevance to state it "regulates nucleocapsid curvature".

(2) "Hyper-closed SeV clam-shaped structure". "Double-headed SeV nucleocapsid". This a completely artificial observation depending on the buffer conditions. As cited in [29], similar preparation of MeV nucleocapsid like particles does not show such assemblies, and in many other cases. The nucleocapsid isolated from MeV virions does not show the same structure either (Virology. 2002 May 10;296(2):300-7). There is no biological relevance for such an observation. "clam-shaped structure". This term is a strange use of the word "clam". The same term was used by this group for their Newcastle Disease Virus (NDV) work. Apparently, their "clam-shaped structure" refers to the assembly of two spiral rings on top of each other. How such a complex resembles a clam is beyond my imagination. The outline of the "clam" in Fig. 1e is arbitrary and random. "NWT is mostly resistant to the second cut, indicating that the clam-shaped structures in NWT will provide extra protection against further digestion and help stabilize nucleocapsids." During virus infection, NP protein is not subject to protease-digestion, let alone "the second cut". "help 'to' stabilize nucleocapsids" is not needed.

(3) "assembly". "our structural data support the occurrence of a previously unreported additional interface between neighboring protomers." Since direct structure superposition with the N-RNA complex of MeV is not presented (perhaps purposely), such a claim is not substantiated. "we observed that the SeV nucleoprotein has an extended loop (Loop20-46) connecting the N-arm and the core of NTD (Fig. 1b)." Again, this strand is named a loop strangely. By visual inspection, the structure in this region is identical to that in MeV NP. "Loop20-46, along with a loop from CTD (Loop312-320) and a loop from the NTD (Loop92-102) assembles into a closed ring". What ring? According to Webster's, "ring" is a circular line. Where is the ring? Again, the structure in this area seems to be the same in MeV NP. All these issues are resolved easily by direct superposition of the two structures. "prevents nucleoproteins from peeling off the nucleocapsids". No such a thing. Removal of RNA by RNase left an assembly of the nucleocapsid proteins almost the same as the nucleocapsid. Read published literature.

(4) "Discussion". "it will be interesting to explore both the occurrence of such double headed structures throughout this order and to examine the specific cellular conditions in which this nucleocapsid type is preferentially formed." From the vast amount of publications in the field, this artificial assembly is only reported by this group under their in vitro conditions. In study of nucleocapsid-like particles of NSVs derived from recombinant proteins, various assemblies may be observed depending on study conditions. Reports on artificial structures are irrelevant to biology. The only nucleocapsid structure that is biologically interesting is the nucleocapsid of the full length viral genome. Fig. 5 caption. "in one SeV nucleocapsid are highlighted". Such a structure does not exist in one nucleocapsid even in the "clam-shaped" structure perpetuated by the authors. It would require a second "nucleocapsid". The genome in a single nucleocapsid is fully protected without the so called "clam-shaped" structure.

Science advancement is brought forward by discoveries of all scientists in the field. The published literature constituted the knowledge base and nomenclature in the field of studying NSV nucleocapsids. The erroneous statements by the authors are clearly falsified by published data from a large number of research labs in the field. The authors are advised to read the published literature and get their basic knowledge in the field straight before making unsubstantiated statements.

Reviewer #3 (Remarks to the Author):

The manuscript titled "Structure and assembly of double-headed Sendai virus nucleocapsids" by Zhang and colleagues reports the structural details of specific components in double headed Sendai Virus nucleocapsids using cryoelectron microscopy. The authors found that expression of the nucleocapsid protein, in multiple expression systems, resulted in a large proportion of double-headed helical stems connected by clam-shaped joints. The cryoEM structures of the helical and clam-shaped regions were separately resolved to 2.9 Å and 3.9 Å respectively, and atomic models of the protein and RNA components could be fitted in satisfactorily. This group had previously

reported the clam-like structure formed by the nucleoprotein of Newcastle Disease Virus. The clam-shaped joint from SeV was found to have a hyper-closed interface as opposed to the NDV analogue. Further, the authors identified additional regions in the nucleocapsid protein that contribute to interfaces and established the role of the C-terminal tail in controlling the curvature of the particles.

The study is meticulous and the results will add substantially to the literature on nucleocapsid assembly of mononegaviruses. It is therefore important to place this study in context with previously published structural analyses, which the Introduction fails to do. The Introduction should be expanded to explain at least the previous work on NDV in more detail. This will make the current results easier to follow.

Other details like differences in primary sequence and nucleocapsid expression conditions should be mentioned. While the NDV nucleocapsids primarily formed clam shaped joints upon recombinant expression, while the SeV protein primarily formed filaments. Was this due to differences in expression conditions or inherent in the proteins? What is the sequence identity among nucleocapsid proteins in mononegaviruses?

Likewise, what is the biological significance of the hyperclosed interface in SeV vs the relatively open area in NDV? Does this point to any difference in the transcription modalities of the two viruses, or are these different states of assembly, to be expected in all members of the family?

How does curvature affect gene transcription? The removal of the C-tail results in the formation of much straighter structures, which is quite obvious from Figures 3 c,d. Is it possible to quantify the degree of stiffness or flexibility of the stems by calculating persistence length, as was done for tropomyosin fibres?

What happens to assembly upon deletion of the N-ring or loop 240-248? Is there any existing information from any other mononegavirus?

The authors mention that in the clam structure, the density for protomers further away from the centre are more resolved, and these were masked and averaged to improve resolution. Is it possible that the protomers do have minute differences in conformation, based on their orientation in context of the helical stems, which was averaged out in the process? Could such slight conformational differences exist in protomers, and could it be of structural or functional significance?

Minor comment:

The colour distinctions are unclear in panel a of Figure 5.

Reviewers' comments:

Reviewer #1 (Remarks to the Author):

In the presented manuscript, Zhang et al present the reconstruction of the Sendai Virus N-RNA complex at atomic resolution, including the helical part and a clam structure previously described by the same authors for NDV. Also, they demonstrate the presence of an additional, ring like structural motive stabilising N-protein interactions. In general, the manuscript is well written and the figures are clear and easy to understand. Also, the manuscript is accompanied by a substantial amount of supplementary information and movies to better describe the findings for the interested reader.

However, in the opinion of this reviewer, some points should be addressed before the manuscript can be considered for publication.

A) In the introduction, please mention that SeV is part of the Paramyxoviridae and indicate which structures have already been solved of the N-RNA complex in this virus family and how different they are at the aa versus the structural level.

Re:

Thanks for reviewing our manuscript.

We have emphasized the belonging of SeV to the family of *Paramyxoviridae*, listed some representative structures of N-RNA complexes resolved before, and mentioned the conservation of nucleoproteins in both sequence and structural levels in the revised manuscript (see *line 34-53*). Thanks for the great suggestion.

B) In the section line 64-72, you talk about the complex purified from HEK cells. However, as far as I understand, all the data for further structural and functional analysis, were generated employing protein expressed in bacteria. I find this confusing, as it makes it more difficult for the reader to follow your line of thought and immediately raises the question why you didn't work with the protein expressed in eukaryotic cells. So please either leave the data on the HEK cells out or explain / comment / discuss the approach you chose.

Re:

Thanks for the great suggestion.

We have moved the data from HEK cells to the “Discussion” in the revised manuscript to avoid possible confusion (see *line 229-233*). This result verifies that the formation of double-headed nucleocapsids is independent on the expression systems (HEK cells and *E. coli*) and the purification strategies (ultracentrifugation and tandem affinity and size exclusion chromatography).

After we obtained SeV nucleoproteins from HEK cells, we performed PTM (posttranslational modifications) analysis via Mass Spectrum, and didn't find PTM on nucleoproteins purified

from HEK cells (unpublished data). Thus, we resolved all the high-resolution structures on nucleoproteins from *E. coli* due to the large yield.

C) line 73-77 are difficult to understand. Please rephrase

Re:

Thanks for the suggestion.

We have deleted the confusing description on helical stems of double-headed SeV nucleocapsids and rephrased the whole paragraph (see *line 84-86*).

D) line 81-84: please explain how cleavage was achieved (trypsin?, which is mentioned later in the manuscript) and where approximately cleavage took place. If C-tail refers to specific aa, please indicate them here.

Re:

Thanks for the question.

Residual impurity in purified proteins could cut off the C-terminal tail (N-tail) over a long period (usually several weeks, called as the aged nucleocapsid). We just noticed that this approach had been adopted in yielding straight nucleocapsids in mumps virus by Ming Luo's group (*Cox et al., Journal of Virology 2013, Severin et al., Journal of Virology 2016*). We have cited these papers in the revised manuscript.

Due to the intrinsically structural flexibility of N-tail, N-tail is susceptible to the residual impurity (see *line 90-95*). Based on our SDS-PAGE gel and high-resolution structures (residue coverage from 3 to 414), the cleavage site of the aged nucleocapsids occurs after residue 414 of N-tail. In our later trypsin cleavage assay, leftover bands are similar in molecular weight to the aged nucleocapsids. Considering that there are 11 R/K residues in N-tail, trace trypsin or other proteases in the purified proteins might take role in the removal of partial N-tail.

E) line 94ff: the reconstruction of the clam structure to my understanding was performed on ring like structures. Please explain in the text how you validate that the structure observed between helical stems is indeed the same as the rings?

Re:

This is really a good point.

Single particle analysis was performed on ring-like particles to reconstruct SeV clam-shaped structure. Unfortunately, ring-like particles have strong preferred orientation, and only top-on views are visible after 2D classification (P2P_Figure 1). Just as what we mentioned in the methods, we manually picked up the clam-shaped joints (side-view) between helical stems and merged them with ring-like particle set. The merged particle set were used to shoot the high-resolution clam-shaped structure. After the local refinement, we obtained a 3.9 Å of resolution cryo-EM map. Considering that joints between double-headed nucleocapsids were utilized in the final 3D reconstruction, the clam-shaped joints between helical stems should

be as same as clam-shaped structures. This is also verified by the similar helical parameters between helical stems and single-turn spiral in clam-shaped structure (see *line 120-124*). We have added more details into the revised manuscript (see *line 106-111*).

P2P_Figure 1. Top-on views only for ring-like particles after 2D classification

F) line 100ff: As I am not sure I understood correctly, so the reconstruction strategy was to use 5 protomers (please indicate what protomers further away from the gap refers to, especially indicate what the gap is) and then use the circular symmetry and turn them one monomer further? Please explain in more detail so that it is easier for the reader to understand.

Re:

Thanks for the good suggestion.

The top-on view of SeV clam-shaped structure highlights its crescent shape, in which the protomers further away from the “crescent gap” are resolved better than those closer to the gap. We labeled the gap and the 5 protomers furthest from the gap in *Supplementary Fig. 6d*.

Just as what the reviewer mentioned, we made full use of the local circular symmetry and turned the particle center one protomer further via our own script to expand our dataset (see *line 123-135* in *Supplementary Materials*). We have followed the advice and add more details into the “Methods” to make it easier for the readers.

G) In the paragraph starting line 112, please include some structural comparison to already resolved nucleoprotein structures from for example Measles and NDV to highlight similarities and differences.

Re:

Thanks for the great suggestions.

We have followed the advice and mentioned the structural similarities of nucleoproteins among many mononegaviruses (see *line 45-53*). Meanwhile, we emphasized the previously unreported interface between N-hole and Loop₂₄₀₋₂₄₈, which is shared by SeV, MeV and RSV (see *line 149-154*). Thanks.

H) line 174: To my understanding, this analysis is a quantification, but not a statistical analysis.

Re:

Thanks for pointing out our error.

We have corrected the statistical analysis to semi-quantification analysis in the revised manuscript (see *line 185-187*). Thanks.

I) line 197ff: Was this analysis performed in material purified from HEK cells or bacteria?

Re:

Thanks for the question.

Structural analysis on both helical stems and clam-shaped structures was performed on nucleoproteins purified from *E. coli*, and the statement that Loop₁₁₈₋₁₂₇ seems only involved in the maintenance of clam-shaped joints between two helical stems was also derived from structures obtained from *E. coli*.

To avoid confusion, we followed the advice and moved the visualization of nucleoproteins from HEK cells to the “Discussion”, which is used to verify the occurrence of double-headed nucleocapsids in mammalian cells (see *line 229-233*).

J) line 205ff: The mentioned FXXR/K motive is only conserved in SeV, NDV and MeV in the aligned sequences. Otherwise, it is a R/K motive, at least to my understanding. Please explain further or correct this statement.

Re:

Thanks for pointing out this issue.

From our high-resolution structure, we can clearly locate Loop₁₁₈₋₁₂₇, which contributes to the assembly of clam-shaped structure in both SeV and NDV. Sequence alignment on nucleoproteins among the family of *Paramyxoviridae* exhibits a hydrophobic residue “F/P/M” followed by a positive-charged residue “R/K” in Loop₁₁₈₋₁₂₇, highly conserved in several viruses including SeV, NDV, PIV5, and Nipah virus. Even though F₁₁₈ is not strictly conserved in Nipah virus and Mumps virus, our mutation screening showed that F₁₁₈ is critical for SeV clam-shaped assembly. Actually, Nipah virus has another hydrophobic residue M₁₁₈ followed by a R₁₂₁, and is reported to assemble into a clam-shaped structure, too (*De-Sheng Ker, bioRxiv 2020*).

Based on these results, we accepted the advice and modified the motif-related statement (see *line 214-219*).

K) line 225ff: Please mention in which members of the Mononegavirales you found these clam structures and in which context (protein purified from over expression / purified from viruses or virus infected cells / visualised directly in the virus particle)

Re:

Thanks for the suggestion.

We have followed the advice and added the details about the clam-shaped structures in the revised “Discussion” (see *line 222-233*).

Briefly, we over-expressed SeV nucleoproteins in *E. coli* and found clam-shaped structures under cryo-EM. Similar clam-shaped assemblies were also found in NDV (Xiyong Song *et al.*, *eLife* 2019) and Nipah virus (De-Sheng Ker, *bioRxiv* 2020) under similar expression and purification conditions (tandem affinity and size exclusion chromatography). Meanwhile, SeV nucleoproteins were also overexpressed in HEK293F mammalian cells and isolated with a two-step ultracentrifugation method, very similar clam-shaped structures were visualized under cryo-EM, too (*Supplementary Fig. 12a*). Given the common occurrence of clam-shaped structures in multiple viruses under different conditions, clam-shaped structures might be popular in the family of *Paramyxoviridae*.

Furthermore, we attempted to verify the occurrence of clam-shaped structures in SeV virion. Even though that the number of clam-shaped structures might be limited to 1 or 2 per virion, we have captured some clam-shaped structures surrounded by two typical herringbone structures packing in a tail-to-tail mode (*Supplementary Fig. 12b*). We are still working hard to collect more tomograms and shoot high-resolution structures of clam-shaped structures *in situ*.

L) Discussion: The finding of double headed nucleocapsids for viruses not encoding a segmented genome is intriguing. Could you please further elaborate on the possible function, what your underlying evidence is to propose the model shown in Fig 5, whether these structures are present in virus particles and virus infected cells and how the occurrence of these structures could be affected by the production system / purification procedure?

Re:

This is really a great question.

1. SeV clam-shaped structure: The ability for non-segmented virus to assemble into a double-headed nucleocapsid is also a surprise to us. Just as what we mentioned before, the occurrence of double-headed nucleocapsids is quite common among the family of *Paramyxoviridae*, which is irrelevant to expression systems (*E. coli* or HEK293F) and purification procedure (tandem binding affinity and size exclusion chromatography, or ultracentrifugation). Intriguingly, we did capture clam-shaped structures surrounded by two typical herringbone structures packing in a tail-to-tail mode after the removal of virion envelope with Triton X-100 (*Supplementary Fig. 12b*). Thus, clam-shaped structures are quite popular and not affected by the production system or purification procedure. Clam-shaped structures and the derived double-headed nucleocapsids might function as the hub to connect two RNA strands in SeV virion with multiple genome copies. Meanwhile, clam-shaped structure is proven more resistant to proteolytic cleavage via self-capping in NDV (Song *et al.* *eLife* 2019).

2. Curved SeV nucleocapsids: Purified SeV double-headed nucleocapsids are usually curved. Trypsin cleavage is a common approach to straighten nucleocapsids for high-resolution structural analysis in many paramyxoviruses. Our high-resolution results provide direct evidence that N-tail is involved in the curvature maintenance. Meanwhile, the curved nucleocapsids also occur insides of the virion. The averaged diameter of SeV virion is ~200 nm, while the total length of its nucleocapsid reaches up to 800 nm. SeV nucleocapsid has to

fold itself several times before packing into the virion. The published (Loney *et al.*, *Journal of Virology* 2009) and our unpublished tomography data shows that SeV usually has 6-8 folds in virion (P2P_Figure 2). Between two neighboring fragments, the joint areas are highly curved or even bent. The exposed N-tail might be easily recognized by phosphoprotein and RdRp to fulfil the genome transcription and replication.

3. Straight SeV nucleocapsids: Accompanied with curved fragments, the straight SeV nucleocapsids are visible in both purified system and in virion. Comparing to the curved SeV nucleocapsids, the straight nucleocapsid has much smaller helical rise and is believed to condensate RNA genome into the virion.

Details have been shown in the manuscript to elaborate the possible function (see *line* 245-256). Thanks for the advice.

P2P_Figure 2. Curved or bent nucleocapsids in SeV virion (Loney *et al.* *Journal of Virology* 2009)

Reviewer #2 (Remarks to the Author):

This manuscript reports the structural analyses of the nucleoprotein (NP) from Sendai virus (SeV), a negative strand RNA virus (NSV), by cryo EM image reconstruction. The coding sequence for the NP has been used to construct an expression vector to express a recombinant protein in HEK293F cells or E. coli. Purification by ultracentrifugation produced assemblies of the NP in complex with cellular RNA, which may have structural features representing the viral nucleocapsid. By application of cryo EM image reconstruction techniques, several structures were determined at different resolutions (However, it is not clear which structure is derived from which protein assembly. The authors randomly use the same term to describe different structures). Primarily, two structures are reported: a helical assembly, and a complex of two spiral rings.

Re:

Thank so much for reviewing our manuscript.

We have followed the advice from the reviewers and moved nucleoprotein purification from HEK293F to “Discussion” to verify that the occurrence of double-headed nucleocapsids is not related to expression system. Thus, the structures described in the main text are all from *E. coli*, which is much clearer than before (see *line 229-233*). Thanks.

The reported structure model was built and refined starting from the coordinates of measles virus (MeV) nucleoprotein, a closely homologous structure. By visual inspection, the reported structure of the SeV helical complex of NP-RNA is nearly identical to that of MeV. Without direct comparisons between the two structures, it is not clear what new features are discovered in SeV nucleocapsid.

Re:

Thanks for the great question.

The helical measles virus nucleocapsids resolved by Schoehn’s lab (*Gustche et al., Science 2015*) provides the first high-resolution (4.3 Å) helical structures and is one of landmark papers in the mononegavirus field. Considering that the sequence identity/similarity between MeV and SeV is ~27%/45%, we mimicked a pseudoatomic model of SeV derived from MeV atomic model. We docked SeV pseudoatomic model into our cryo-EM maps, followed by a flexible fitting. 2.9 Å resolution of SeV nucleoproteins allows us to build the model more accurately. In the overall structure, SeV nucleoprotein has a similar shape with NDV, which consists of NTD and CTD with the RNA cleft between. These structural features are also conserved among the family of *Paramyxoviridae*. The RMSD of nucleoproteins between MeV and SeV is 1.4 Å, and only some loops exhibit obvious difference.

Actually, our manuscript focuses on a double-headed SeV nucleocapsid, which has not been discovered in helical MeV nucleocapsid. The formation of double-headed SeV nucleocapsid is mainly dependent on the loops from opposite nucleoproteins, and such structural feature seems not shared by MeV nucleocapsid.

Furthermore, in helical stems of SeV double-headed nucleocapsids, the paired N-hole and Loop₂₄₀₋₂₄₈ functions as an anchoring site to connect two neighboring subunits. Even though

there are similar structures in MeV and NDV, these interfaces and the roles of N-hole and Loop₂₄₀₋₂₄₈ are not paid too much attention and not reported in previous literatures.

In all, SeV clam-shaped structures and the new interface between N-hole and Loop₂₄₀₋₂₄₈ are the two new structural features, which are not disclosed from MeV.

However, there are too many incorrect statements in the manuscript that are inconsistent with published data and the established nomenclature/knowledge in the field. Without careful revision and thorough literature citation, this manuscript does not have the quality to be published in a respectable scientific journal. The authors are advised to read published literature thoroughly to have a good understanding of the field and the subject in discussion. Without such knowledge, the presentation by the authors is scientifically weak, or even useless.

(1) "tail". Sendai virus is a prototype NSV that has been studied for many decades in the field. A set of nomenclature has been well established to describe the structure and function of SeV NP. The domain at the C-terminus of SeV NP has been named N-tail (residues 402-524). It is completely foolish for authors to rename it C-tail. In the section named "C-tail regulates nucleocapsid curvature" by the authors, erroneous statements are numerous. e.g. "the C-tail of mononegaviruses is known to function in the regulation of gene replication and transcription via binding to phosphoprotein^{3,23-26}." The works cited in this sentence did not use a nucleocapsid protein that has a N-tail. In addition, not all viruses in Mononegavirales have a NP that contains a N-tail.

Re:

Thank so much for the criticism.

It is not wise for us to rename N-tail to C-tail based on its position at the C-terminus. We accepted the criticism and corrected all of them in the revised manuscript.

We also double checked all the citations, and tried to avoid incorrect ones. Thanks so much.

e.g. "These 12 residues assemble into a loop (Loop₄₀₃₋₄₁₄)". It is not understandable why the authors named this strand "a loop". More critically, the conformation of this strand in a N-tail removed protein may be totally different from that in the full length protein.

Re:

Thank the reviewer for raising the issue about Loop₄₀₃₋₄₁₄.

We used the algorithm implemented in UCSF Chimera (*Kabsch et al., Biopolymer 1983*) to define the secondary structure for regions from 403 to 414, and this region is unstructured and is classified as coils. Usually, unstructured regions found between regular secondary structural elements (α -helix and β -strand) are named as loops. Considering that there might be some ordered structures after the residue 414, we prefer to labeling the region from 403-414 as Loop₄₀₃₋₄₁₄.

Full-length paramyxovirus nucleoproteins are usually curved, which is hard to shoot high-resolution structures. Usually, protease treatment to cleave N-tail from full-length nucleoproteins is widely used to yield straight filaments for high-resolution analysis. To our

knowledge, N-tail including the start region from 403-414 has not been resolved in almost all ring-like or helical nucleocapsids due to its intrinsic flexibility before our work on SeV nucleoproteins. It will be great to compare structures between SeV NC_{cleaved} with Loop₄₀₃₋₄₁₄ and full-length NC_{WT} resolved via other approaches in the future. Thanks.

“our data provide direct evidence that the “C”-tail extends outside of the nucleocapsids (Fig. 3a, b)”. However, published data showed that the N-tail of the full length NP may be inside or outside the helical nucleocapsid.

Re:

Thanks for the question.

We have rephrased this sentence and limited our statement to SeV nucleocapsid (see *line 156-157*). Thanks.

Briefly, in the helical stem of SeV nucleocapsid, Loop₄₀₃₋₄₁₄, as the start region of N-tail, points to the outside of helical nucleocapsid. What is more, the direct contact between neighboring rungs in the helical nucleocapsid is so tight that will not allow N-tail turning back to the insides. Based on these, we claim that N-tail might extends outside of SeV helical nucleocapsid.

“Loop403-414 to extend about 36 Å from the C-arm, where it reaches the RNA cleft between the NTD and CTD”. The grammar is incorrect. This reaching to the RNA cleft would not occur when the N-tail is intact.

Re:

Thanks for the criticism.

In SeV NC_{cleaved}, Loop₄₀₃₋₄₁₄ extends about 36 Å from the C-arm in our high-resolution structure. Since the missing of the structural information of full-length SeV nucleocapsid, it is hard to speculate whether Loop₄₀₃₋₄₁₄ will reach the RNA cleft between NTD and CTD in the intact SeV nucleocapsids. Interestingly, in Ebola virus nucleocapsid, there is an extended α -helix from the disordered carboxy-terminal region, which reaches the RNA cleft and clamps RNA inside the RNA cleft (Wan *et al.*, *Nature* 2017). The extended α -helix in Ebola virus is very similar to what we observed in SeV nucleocapsid.

We have followed the reviewer’s advice and rephrased our statement in the revised manuscript (see *line 171-173*).

“The undulating surfaces of CTDs positioned between two neighboring rungs of SeV nucleocapsids force Loop403-414 to extend about 36 Å from the C-arm”. How does a surface “force” a peptide to extend? Where is the energy from? “Loop403-414 functions like a lid to cover RNA cleft, which will offer thorough protection against possible damage to RNA”. How can a strand be a lid? According to Fig. 3a, the position of residues 403-414 is far away from the packaged RNA. “damage to RNA”? What damage? Radiation?

Re:

Thanks so much for the criticism.

We admitted that we overinterpreted the role of Loop₄₀₃₋₄₁₄ in this paragraph. We followed the reviewer’s criticism and deleted this part from the revised manuscript. Thanks.

e.g “which directly indicates that C-tail regulates nucleocapsid curvature.” The curvature of the NSV nucleocapsid is completely dependent on environment, including interactions with other biological molecules, ionic strength, or pH. The artificial curvature observed by the authors is dependent on the presence of N-tail under their study conditions. However, this has no biological relevance to state it “regulates nucleocapsid curvature”.

Re:

Thanks for the criticism.

Curved helical nucleocapsids are quite popular in non-segmented nucleocapsids including SeV, MeV, Mumps virus and others. Due to their heterogeneity, it is hard to shoot high-resolution structures directly. To improve resolution, proteases such as trypsin are introduced to remove the flexible regions (usually N-tail), which yields straight nucleocapsids and helps obtain high-resolution structures. MeV nucleocapsids have been reported with curved filaments (full-length) and straight filaments (with N-tail removed) in the same buffer condition (*Bhella et al. Journal of Molecular Biology 2004; Gutsche et al. Science 2015*) (P2P_Figure 3).

Similar to MeV, we incubated both full-length SeV nucleoproteins (N_{WT}) and N-tail truncation under the exactly same buffer condition, and N-tail truncation is much straighter than N_{WT}. Persistence length analysis does show that N_{WT} has a much shorter persistence length at ~228 nm than N-tail truncation (~887 nm). Accordingly, N_{WT} has a much higher bending angle comparing to N-tail truncation.

We accepted the criticism from the reviewer and rephased our statement to “N-tail correlates with SeV nucleocapsid curvature”.

P2P_Figure 3. The structural comparison between Measles virus full-length nucleoprotein and N-tail truncation (*Gutsche et al. Science 2015*)

(2) “Hyper-closed SeV clam-shaped structure”. “Double-headed SeV nucleocapsid”. This a completely artificial observation depending on the buffer conditions. As cited in [29], similar preparation of MeV nucleocapsid like particles does not show such assemblies, and in many other cases. The nucleocapsid isolated from MeV virions does not show the same structure either (*Virology*. 2002 May 10;296(2):300-7). There is no biological relevance for such an observation.

Re:

Thanks for the challenging on the popularity of clam-shaped structures.

Just as the reviewer mentioned that some viruses such as MeV and Mumps virus do not show clam-shaped assemblies. However, since the first discovery of clam-shaped structure from NDV (*Song et al., eLife 2019*), clam-shaped structures and the derived double-headed nucleocapsids have been found in SeV, no matter purified from *E. coli* or HEK293F under different purification and buffer conditions. Very recently, clam-shaped structure was proved existing in Nipah virus, too (*De-Sheng Ker, bioRxiv 2020*). When we revisited the previously published papers, similar clam-shaped structures have been shown with typical double herringbone patten from Hendra virus (as shown in Figure 5, *Pearce et al., Protein Expression and Purification 2015*). Unfortunately, no enough attention was paid on clam-shaped structures at that time. Besides expression and purification of nucleoproteins, we also isolated nucleocapsids directly from NDV and SeV virions and did find some clam-shaped structures (*Supplementary Fig. 12b*) (*Song et al. eLife 2019*) (P2P_Figure 4).

Based on our current results, clam-shaped structures are proposed to provide protection to nucleocapsids via self-capping (*Song et al. eLife 2019*) and benefit polyploid genome organization and genome condensation. As a previously unidentified structure, we are providing more structural evidence for clam-shaped structures and still exploring new functions that clam-shaped structures might have.

	Sendai virus	Newcastle disease virus	Nipah virus	Hendra virus
Representative image				Expression system	Eukaryotic system	Prokaryotic system		
Purification method	Density gradient ultracentrifugation	Nickel affinity column & Gel filtration		
pH value	7.4	7.4&8.0	8.0	7.5

P2P_Figure 4. Summary of clam-shaped structures from different viruses under different conditions

“clam-shaped structure”. This term is a strange use of the word “clam”. The same term was used by this group for their Newcastle Disease Virus (NDV) work. Apparently, their “clam-shaped structure” refers to the assembly of two spiral rings on top of each other. How such a complex resembles a clam is beyond my imagination. The outline of the “clam” in Fig. 1e is arbitrary and random.

Re:

Thanks for the criticism.

Clam-shaped structure refers to two single-turn spirals coating each other via a tail-to-tail manner. The side view of the clam-shaped structure is easily recognized, with two shell-like densities. Based on the shape, we used “clam” in our papers to describe this previously unidentified structure. Actually, before we submitted our NDV clam to *eLife* (Song *et al. eLife* 2019), we consulted several seniors especially from US and they reached an agreement to use the term of “clam” to describe it. To be honest, we haven’t found better terms to name it so far.

Interestingly, clam-shaped structures can further grow into double-headed nucleocapsids. Under cryo-EM, helical nucleocapsid exhibits herringbone-like structures (P2P_Figure 5). Comparing to single-headed nucleocapsid, double-headed nucleocapsids have two herringbone-like structures packing in a tail-to-tail manner, which has distinct features. Based on this, we can recognize and outline the clam joint between two helical nucleocapsids easily and accurately. During our single particle analysis on clam-shaped structures, we also manually picked up the clam-shaped structures in double-headed nucleocapsids as the side views to make up the missing cone issue. Our final 3D reconstruction converges very well and verifies the accuracy of our manual picking on clam joints between two helical nucleocapsids.

P2P_Figure 5. Structural difference between clam-shaped structure, single-headed nucleocapsid and double-headed nucleocapsid

“NWT is mostly resistant to the second cut, indicating that the clam-shaped structures in NWT will provide extra protection against further digestion and help stabilize nucleocapsids.” During virus infection, NP protein is not subject to protease-digestion, let alone “the second cut”. “help ‘to’ stabilize nucleocapsids” is not needed.

Re:

Thanks for the criticism.

We admitted that we overinterpreted our *in vitro* data on protease analysis. We have taken the advice and removed the improper statement. Thanks so much.

(3) “assembly”. “our structural data support the occurrence of a previously unreported additional interface between neighboring protomers.” Since direct structure superposition with the N-RNA complex of MeV is not presented (perhaps purposely), such a claim is not substantiated. “we observed that the SeV nucleoprotein has an extended loop (Loop20-46) connecting the N-arm and the core of NTD (Fig. 1b).” Again, this strand is named a loop strangely. By visual inspection, the structure in this region is identical to that in MeV NP. “Loop20-46, along with a loop from CTD (Loop312-320) and a loop from the NTD (Loop92-102) assembles into a closed ring”. What ring? According to Webster’s, “ring” is a circular line. Where is the ring? Again, the structure in this area seems to be the same in MeV NP. All these issues are resolved easily by direct superposition of the two structures. “prevents nucleoproteins from peeling off the nucleocapsids”. No such a thing. Removal of RNA by RNase left an assembly of the nucleocapsid proteins almost the same as the nucleocapsid. Read published literature.

Re:

Thanks for the criticism.

We took the reviewer’s advice and changed the “N-Ring” to “N-Hole” to more accurately describe the structure formed by Loop₂₀₋₄₆, Loop₃₁₂₋₃₂₀ and Loop₉₂₋₁₀₂.

Near-atomic resolution structure of the helical measles virus nucleocapsid from Schoehn’s lab (*Gutsche et al., Science 2015*) is a landmark work in the paramyxovirus field, which reported the first high-resolution structures of helical nucleocapsid. The authors mentioned that MeV N_{core} oligomerization is mainly mediated via the exchange subdomains of the NTD arm and the CTD arm. Detailly, the NTD arm of the N_i protomer inserts into a groove in the CTD of N_{i+1} subunit where CTD arm lies on top of the N-CTD of the N_{i-1} subunit, generating a repeated helical structure. Interestingly, in MeV structure (as shown in Figure 2A, C, *Gutsche et al., Science 2015*), the N-hole formed by Loop₁₆₋₃₅, along with a helix from CTD (Helix₃₁₇₋₃₂₂) and a loop from the NTD (Loop₉₂₋₁₀₀) is visible (P2P_Figure 6). In MeV, the counterpart Loop₂₄₀₋₂₄₈, which inserts into N-hole, is not clearly visualized. Thus, this new interface was not mentioned in Schoehn’s paper (*Gutsche et al., Science 2015*). Actually, such “hole” structures are quite popular in paramyxoviruses such as RSV (*Tawar et al., Science 2009*), Pneumovirus (*Renner et al., eLife 2016*) and parainfluenza virus 5 (*Alayyoubi et al., PNAS 2015*).

In SeV nucleocapsid, Loop₂₄₀₋₂₄₈ and N-hole were clearly visualized. Based on this, we proposed that the budged Loop₂₄₀₋₂₄₈ and N-hole might play an important role in the assembly. To test this, we followed the reviewer’s advice and did replacement mutation on Loop₂₄₀₋₂₄₈. Very interestingly, there are many stretched thin filaments in the mutant under cryo-EM

(Figure 2d), which probably be caused by the loss of anchoring function. All our results manifest the important roles of Loop₂₄₀₋₂₄₈ in the assembly of helical nucleocapsid.

We also followed the reviewer's suggestion and removed the statement of RNA in the nucleocapsid assembly. Thanks.

P2P_Figure 6. Swapped N-hole and Loop₂₄₂₋₂₄₇ in MeV nucleocapsid

(4) “Discussion”. “it will be interesting to explore both the occurrence of such double headed structures throughout this order and to examine the specific cellular conditions in which this nucleocapsid type is preferentially formed.” From the vast amount of publications in the field, this artificial assembly is only reported by this group under their in vitro conditions. In study of nucleocapsid-like particles of NSVs derived from recombinant proteins, various assemblies may be observed depending on study conditions. Reports on artificial structures are irrelevant to biology. The only nucleocapsid structure that is biologically interesting is the nucleocapsid of the full length viral genome. Fig. 5 caption. “in one SeV nucleocapsid are highlighted”. Such a structure does not exist in one nucleocapsid even in the “clam-shaped” structure perpetuated by the authors. It would require a second “nucleocapsid”. The genome in a single nucleocapsid is fully protected without the so called “clam-shaped” structure. Science advancement is brought forward by discoveries of all scientists in the field. The published literature constituted the knowledge base and nomenclature in the field of studying NSV nucleocapsids. The erroneous statements by the authors are clearly falsified by published data from a large number of research labs in the field. The authors are advised to read the published literature and get their basic knowledge in the field straight before making unsubstantiated statements.

Re:

Thanks for the criticism.

Just as what we mentioned before, clam-shaped structures are not only reported by our labs on the analysis of NDV and SeV nucleocapsids, but also are discovered by Antson's lab in Nipah virus (*De-Sheng Ker, bioRxiv 2020*). Meanwhile, Hendra virus was found with similar clam-shaped structures (*Lesley Pearce et al., Protein Expression and Purification 2015*). So far, it is hard for us to claim that clam-shaped structures are everywhere, but it will be interesting to further explore whether such double-headed nucleocapsids will occur throughout the family of *Paramyxoviridae*, and what role such double-headed nucleocapsids will play.

We totally agree with the reviewer that the most biologically interesting nucleocapsid structure is the nucleocapsid of the full-length viral genome. Cryo-ET might be one of the best solutions to resolve nucleocapsids under native environments. However, limited by the technique, it is still hard to shoot high-resolution structures on the assembly of nucleocapsids via cryo-ET. So, almost all nucleocapsid structures are resolved in rings or helical filaments (not as natural as in virion) after the purification. These high-resolution structures are helpful to deepen the understanding of the whole field.

We have followed the great suggestions from the reviewer and tuned down our voice on the related statements. Thanks so much.

Reviewer #3 (Remarks to the Author):

The manuscript titled “Structure and assembly of double-headed Sendai virus nucleocapsids” by Zhang and colleagues reports the structural details of specific components in double headed Sendai Virus nucleocapsids using cryoelectron microscopy. The authors found that expression of the nucleocapsid protein, in multiple expression systems, resulted in a large proportion of double-headed helical stems connected by clam-shaped joints. The cryoEM structures of the helical and clam-shaped regions were separately resolved to 2.9 Å and 3.9 Å respectively, and atomic models of the protein and RNA components could be fitted in satisfactorily. This group had previously reported the clam-like structure formed by the nucleoprotein of Newcastle Disease Virus. The clam-shaped joint from SeV was found to have a hyper-closed interface as opposed to the NDV analogue. Further, the authors identified additional regions in the nucleocapsid protein that contribute to interfaces and established the role of the C-terminal tail in controlling the curvature of the particles.

The study is meticulous and the results will add substantially to the literature on nucleocapsid assembly of mononegaviruses. It is therefore important to place this study in context with previously published structural analyses, which the Introduction fails to do. The Introduction should be expanded to explain at least the previous work on NDV in more detail. This will make the current results easier to follow.

Re:

Thanks for reviewing our manuscript and the great suggestions.

In the revised manuscript, we have introduced more about previously published structures among paramyxoviruses, and emphasized the clam-shaped structures formed by NDV and its structural peculiarity (see *line 45-63*). Thanks.

Other details like differences in primary sequence and nucleocapsid expression conditions should be mentioned. While the NDV nucleocapsids primarily formed clam shaped joints upon recombinant expression, while the SeV protein primarily formed filaments. Was this due to differences in expression conditions or inherent in the proteins? What is the sequence identity among nucleocapsid proteins in mononegaviruses?

Re:

Thanks for the good suggestion.

Briefly, NDV and SeV are purified from *E. coli* under the same expression conditions and the same purification strategy, except that there is an extra sucrose ultracentrifugation step to further isolate NDV clam-shaped nucleoproteins. In our *eLife* paper on NDV nucleoprotein, to reconstruct the previously unidentified clam-shaped structures, we had to obtain pure fractions of clam-shaped particles with extra ultracentrifugation. In SeV, we wanted to make one step forward and aimed to resolve the whole double-headed nucleocapsids. So, we skipped the ultracentrifugation step. Actually, in both purified NDV and SeV, besides clam-shaped structures, double-headed filaments in different lengths are visible in the early

fraction of gel filtration. Thus, we believe there might be no obvious difference in the distribution of clam-shaped structure and helical filaments between NDV and SeV nucleoproteins.

Sequence alignment on nucleoproteins shows that sequence identity among NDV and SeV is only ~27%. Despite of poor sequence conservation, paramyxovirus nucleoproteins exhibit well-conserved architectures. In NDV and SeV, nucleoproteins have conserved hydrophobic residue “F” followed by positive charged residues “K/R”, which is critical for the assembly of clam-shaped structures. Another nucleoprotein from Nipah virus has these conserved residues and has been reported with the similar clam-shaped structure (*De-Sheng Ker, bioRxiv 2020*). A closer look at the published literature on nucleoproteins from Hendra virus have found similar clam-shaped structures (*Lesley Pearce et al., Protein Expression and Purification 2015*). All these results manifested the popularity of clam-shaped structure among paramyxoviruses.

Certainly, we also notice that not all paramyxoviruses have this motif or have clam-shaped structures. For instance, we purified MuV nucleoproteins, following the exactly same purification strategy as SeV. To our great surprise, MuV nucleoprotein doesn’t show clam-shaped structures under cryo-EM. Similar to the previous literatures, full-length MuV nucleoproteins will exhibit ring-like structures after purification (*Cox et al., Journal of Virology 2009*).

Likewise, what is the biological significance of the hyperclosed interface in SeV vs the relatively open area in NDV? Does this point to any difference in the transcription modalities of the two viruses, or are these different states of assembly, to be expected in all members of the family?

Re:

Thanks for the question.

In our previous *eLife* paper (*Song et al., eLife 2019*), clam-shaped structures and the derived double-headed nucleocapsids from NDV are vital for GFP reporter gene in the minigenome assay. In NDV clam-shaped structure, two single-turn spirals are held together via loops from neighboring opposite nucleoproteins. In SeV, the clam-shaped structure exhibits a crisscrossed pattern and the space between two opposite single-turn spirals becomes much smaller. Based on these, we named SeV clam-shaped structure as a hyperclosed state, comparing to NDV clam-shaped structure in a closed state.

The hyperclosed clam joint in SeV renders a closer contact between two single-turn nucleocapsids, which might stabilize the double-headed nucleocapsid. Actually, multiple genomes packing into one enlarged virion (ranging from 110 nm to 540 nm) has been easily recognized and systemically studied in SeV virion (*Loney et al. Journal of Virology 2009*). Interestingly, our unpublished cryo-ET data on NDV virion shows that the diameter of NDV also ranges from 100 nm to 600 nm, but less multiple genome packing is shown (P2P_Figure 7). So far, we have no direct evidence that the difference in clam-shaped structures between NDV and SeV will influence the transcription/replication. This is really a very interesting question to be investigated in the near future.

Just as what we mentioned before, clam-shaped structures are proved prevalent in NDV, SeV, Nipah virus and Hendra virus. Our unpublished data shows that Mumps virus nucleoproteins will not assemble into clam-shaped structures under the same purification condition. Thus, we will not claim that closed or hyperclosed clam-shaped nucleocapsids will exist in all paramyxoviruses.

P2P_Figure 7. Micrographs on intact NDV virion. Empty

How does curvature affect gene transcription? The removal of the C-tail results in the formation of much straighter structures, which is quite obvious from Figures 3 c,d. Is it possible to quantify the degree of stiffness of flexibility of the stems by calculating persistence length, as was done for tropomyosin fibres?

Re:

Thanks for the great suggestion.

Curved nucleocapsids are quite popular in the family of *Paramyxoviridae*. To shoot high-resolution structures, protease cleavage especially on the C-terminal tail (N-tail) is a common way to straighten nucleocapsids. In NDV, the removal of N-tail will influence genome transcription/replication in the minigenome assay (see in Figure 3-figure supplement 1, *Song et al. eLife 2019*). N-tail is reported mostly disordered, but plays important roles via binding phosphoprotein and then recruiting RdRp to initiate gene transcription. The orientation of N-tail is of uncertainty in the family of *Paramyxoviridae*. In SeV, the first 12 residues of N-tail were resolved in our structures, which lies between neighboring rungs. The remaining N-tail will follow the direction, pointing to the outsides of helical nucleocapsids. Disordered N-tail on the outer surface might incur the imbalance and cause curvature to the helical nucleocapsid. Curved nucleocapsids might expose N-tail to a higher extent and be accessed by phosphoprotein and RdRp more easily. Actually, in SeV virion, the whole nucleocapsids usually have several fragments, and the linker regions are highly curved or even bent.

Thanks for the great suggestion to measure the persistence length of nucleocapsids under different conditions. Our results show that the persistence length of nucleocapsids after the removal of N-tail is 228 nm, comparing to 877 nm in wild-type nucleocapsids, which is inserted in the revised manuscript. Obviously, the removal of N-tail results in the formation of much straighter filaments. More interestingly, we do believe persistence analysis can be

measured in nucleocapsids in virion as revealed by cryo-ET, helpful to clarify the nucleocapsid folding and encapsulation into virion. Thanks for the great suggestion.

What happens to assembly upon deletion of the N-ring or loop 240-248? Is there any existing information from any other mononegavirus?

Re:

Thanks for the question.

Besides the swapped N-arm and C-arm, we found that Loop₂₄₀₋₂₄₈ of one protomer inserts into N-hole of the neighboring one, which functions as an anchor site for the assembly of SeV nucleocapsids. To check this, we followed the reviewer's advice and did replacement mutation on Loop₂₄₀₋₂₄₈. Very interestingly, there are many stretched thin filaments in the mutant, which probably be caused by the loss of anchoring function (Figure 2d). These results manifest the important roles of Loop₂₄₀₋₂₄₈ and N-hole in the assembly of helical nucleocapsid.

Actually, such interface between swapped Loop₂₄₀₋₂₄₈ and N-hole does occur in the helical nucleocapsid from Measles virus (*Gustche et al., Science 2015*), which was not paid too much attention. There is a similar loop named Loop₂₄₂₋₂₄₇ in helical structure of Measles virus (PDB ID 4UFT), which inserts into a N-hole like structure. Interestingly, similar interface also occurs in RSV ring structure (PDB ID 2WJ8) and NDV clam-shaped structure (PDB ID 6JC3) (*Tawar et al. Science 2009; Song et al. eLife 2019*). Thus, such interface might be popular among mononegaviruses.

The authors mention that in the clam structure, the density for protomers further away from the centre are more resolved, and these were masked and averaged to improve resolution. Is it possible that the protomers do have minute differences in conformation, based on their orientation in context of the helical stems, which was averaged out in the process? Could such slight conformational differences exist in protomers, and could it be of structural or functional significance?

Re:

Thanks for the great suggestions.

Considering that the number of protomers in SeV clam-shaped structures is not always the same, the heterogeneous samples will yield a cryo-EM map only at the overall resolution of 5.6 Å, and protomers closer to the gap between two single-turn spirals is resolved at even lower resolution. Our local resolution map shows that protomers further away from the center are better resolved. To further improve the resolution, these protomers were masked and averaged to improve resolution to 3.9 Å for accurate model building. Unfortunately, the averaging process ignored the possible minute conformational difference between protomers especially around the gap, which it is really worthy of further investigation in the future.

Considering that clam-shaped structures follow its helical pattern and can further grow into double-headed nucleocapsids, our bold hypothesis is that the protomers in clam-shaped structure might be similar.

Minor comment:

The colour distinctions are unclear in panel a of Figure 5.

Re:

Thanks for the suggestion.

We have increased contrast in the revised Figure 5.

Reviewers' comments:

Reviewer #1 (Remarks to the Author):

I would like to thank the author for their efforts to address the questions raised during the first round of revision. Most concerns have been addressed to my satisfaction. I would kindly ask the authors to consider the following points to improve their manuscript:

- a) Line 193: VSV is not a paramyxovirus
- b) Line 186ff: please add numbers for sequence conservation and also calculate the rmsd values for SDV vs MeV N or other related paramyxoviridae.
- c) Line 368ff: It would be really helpful to have a structural comparison with the N-protein assembly of other paramyxoviridae that exhibit the hole structure, also regarding the surface charge.
- d) Line 515ff: These are really nice data. I would kindly ask you to comment on the length differences observed between the mutants. Is it possible that the reduced number of long filaments is affecting your results (N118)?
- e) Line 549ff: Thank you for providing the tomography data. Could you please elaborate why you used Triton lysis of the virions and whether this could affect the morphology of the RNP? Also, I am unfortunately not able to see the clam-shaped structure in sup fig 12b. Could you please perform segmentation or use different filtering or AI based image enhancement to improve the visualization?
- f) Line 912: I would kindly ask the authors to rephrase the figure legend, as the proposed functions have not been proven. Therefore, I would propose to write 'A model illustrating the proposed / hypothesized distinct functions of different assembly forms'

Reviewer #2 (Remarks to the Author):

The submitted revision keeps the fundamental flaws of the original submission.

1, In authors' rebuttal, they stated "The RMSD of nucleoproteins between MeV and SeV is 1.4 Å", which means the two structures are identical given the experimental conditions. The Fig. 1b in this manuscript showed identical structure as that in Fig. 2A in Gutsche et al. 2015. There is no "newly discovered hole". No one can discover a new structure feature if the structure is identical to a previously published structure, period. The conformation of the truncated C-terminal region in this structure is artificial, having no relevance in the infectious nucleocapsid. In conclusion, this reported structure is identical to that of MeV nucleocapsid.

2, The authors continue to perpetuate biological relevance of "the clam-shaped structure". State here again: The genome of paramyxoviruses is a single stranded negative strand RNA, encapsidated in the nucleocapsid. A single strand genome in a virion has the full functions for virus replication. The authors continue to make erroneous statements: "the number of clam-shaped structures might be limited to 1 or 2 per virion", in Fig. 5, they claim "In one SeV nucleocapsid, clam-shaped joint and straight/condensed filaments coexist with loosed or even uncoiled filaments". Impossible!!! It is simple math, one nucleocapsid cannot have two heads, it is a single strand. The large portion of infectious SeV virions contains a single nucleocapsid and are fully infectious. "these double-headed nucleocapsid structures can confer benefits for genome stability, polyploid genome organization and genome condensation". A single nucleocapsid is completely stable, has normal condensation, and is fully infectious. If the authors believe that this pairing is relevant to multiple nucleocapsids in a virion, they need more data to support this claim, but it is not relevant to the infectivity of paramyxoviruses, especially SeV.

Reviewer #3 (Remarks to the Author):

In the revised manuscript, the authors have addressed all questions posed by this referee. There are no further concerns and the manuscript may be accepted for publication.

Reviewers' comments:

Reviewer #1 (Remarks to the Author):

I would like to thank the author for their efforts to address the questions raised during the first round of revision. Most concerns have been addressed to my satisfaction. I would kindly ask the authors to consider the following points to improve their manuscript:

a) Line 193: VSV is not a paramyxovirus

Re:

Thanks so much for pointing out our error.

We have removed the citation on VSV N⁰P complex from the manuscript (see *line 49-53*).

Thanks.

b) Line 186ff: please add numbers for sequence conservation and also calculate the rmsd values for SDV vs MeV N or other related paramyxoviridae.

Re:

Thanks for the suggestion.

The sequence identity among paramyxovirus nucleoproteins including SeV, NDV, MeV, MuV, PIV5 and NiV is in the range of 24-29% (P2P_Figure 1). Even though not high sequence identity, the nucleoprotein structures among all these paramyxoviruses are quite conserved with the pruned RMSD less than 1.4 Å (see *line 216-217*).

The sequence identity and structural conservation between SeV N and others have been mentioned in the revised manuscript with a new figure (*Supplementary Fig. 12*) (P2P_Figure 1).

P2P_Figure 1. Sequence and structural comparisons of nucleoproteins in the family *Paramyxoviridae*.

c) Line 368ff: It would be really helpful to have a structural comparison with the N-protein assembly of other paramyxoviridae that exhibit the hole structure, also regarding the surface charge.

Re:

Thanks for the great suggestion.

Big assemblies and their respective atomic models including either ring structure, helical structure or clam-shaped structure have been reported in PIV5 (PDB-4XJN), MeV (PDB-4UFT) and NDV (PDB-6JC3), respectively. After displaying three consecutive protomers, we can see clearly the similar N-hole structures in PIV5, MeV and NDV as in SeV (SeV N-hole structure is shown in *Figure 2*). Interestingly, N-holes in these paramyxoviruses are filled with loops from the neighboring protomers, which might anchor neighboring protomers together. Further structural analyses indicate the occurrence of electrostatic interaction between the extended loops and N-holes in these paramyxoviruses (see *line 151-154*).

We have prepared a new figure (*Supplementary Fig. 9*) (P2P_Figure 2) and placed it in the revised manuscript.

Thanks.

P2P_Figure 2. The swapped interface between the extended loops and N-holes in NDV, PIV5 and MeV.

d) Line 515ff: These are really nice data. I would kindly ask you to comment on the length differences observed between the mutants. Is it possible that the reduced number of long filaments is affecting your results (N118)?

Re:

Thanks for the good question.

To screen the key residues for double-headed nucleocapsids, we did replacement mutants on Loop₁₁₈₋₁₂₇, purified mutant proteins and examined them under cryo-EM. Just as what we showed in *Supplementary Fig. 1*, different fractions (0.2 mL per each) in the same peak after gel filtration show nucleocapsids in different lengths and even in different assembly forms. The shorter filaments from N_{F118A} mutant are most probably due to the later fraction we collected for cryo-EM examination.

Indeed, in 40 cryo-EM micrographs we collected for N_{F118A} mutant, we did not find any double-headed nucleocapsids. As the control, N_{F119A} mutant has the similar length as N_{F118A} mutant, but exhibits tons of typical double-headed forms.

For better exhibition, we replaced the original micrograph with the new one, which has longer single-headed nucleocapsids (*Supplementary Fig. 11*) (P2P_Figure 3).

Thanks.

P2P_Figure 3. Micrograph replacement to longer filaments of SeV N_{F118A}. **a** A micrograph with shorter filaments. **b** A new micrograph with longer filaments. No double-headed nucleocapsid is visible in both the original micrograph and the new one.

e) Line 549ff: Thank you for providing the tomography data. Could you please elaborate why you used Triton lysis of the virions and whether this could affect the morphology of the RNP? Also, I am unfortunately not able to see the clam-shaped structure in sup fig 12b. Could you please perform segmentation or use different filtering or AI based image enhancement to improve the visualization?

Re:

Thanks for the great question.

Direct visualization on the intact virion was our first choice to verify the existence of double-headed nucleocapsids *in vivo*. After we isolated SeV virions via ultracentrifugation, we vitrified intact SeV virions and collected tomograms. It is easy for us to recognize the Spike protein on the virion surface after frame alignment (*P2P_Movie 1*). Unfortunately, the nucleocapsids wrap each other and it is quite hard to see detailed structures, which is similar to recent tomography studies on Spike proteins and nucleocapsids of COVID-19 (*Ke et al. Nature 588, 498–502, 2020*).

We suspected that the contents of SeV virion and the wrapped nucleocapsids might hamper the recognition of double-headed forms. Then, we turned to use detergent to lyse virion as an alternative, which could spread out nucleocapsids on cryo-EM grids. To avoid possible recombination of nucleocapsids from different virions during treatments, we used low-concentration (2% v/v) Triton on diluted SeV virions. The mild treatment does spread out filaments and provides possibility for us to recognize clam-shaped structures in double-headed nucleocapsids.

A typical character of double-headed clams under cryo-EM is that two herringbone-like structures are packed in a tail-to-tail manner. After screening, we manually picked up good candidates with double-headed forms (see in *Supplementary Fig. 14*). Limited by the noisy background caused by tons of proteins released from SeV virion, the double-headed nucleocapsids are not as sharp as purified nucleocapsids.

To check the accuracy of our manual picking, we utilized our reconstructed clam-shaped structures as the reference and performed a reference-based particle picking on micrographs. The automatically picked particle with the highest cross-correlation score fits well with our manual picking (*P2P_Figure 4*). Currently, we are doing tomography averaging on intact

SeV virion with large size (> 300 nm), and hope we can provide higher-resolution 3D clam structures of SeV nucleoproteins.

We followed the reviewer's great advice and sharpened our micrographs. Now, they look much better (*Supplementary Fig. 14*).

Thanks.

P2P_Figure 4. Reference-based particle picking on clam-shaped structures from nucleocapsids directly released from SeV virion.

f) Line 912: I would kindly ask the authors to rephrase the figure legend, as the proposed functions have not been proven. Therefore, I would propose to write 'A model illustrating the proposed / hypothesized distinct functions of different assembly forms'

Re:

Thanks for the great suggestion.

We have followed the advice and rephrased the figure legend for Figure 5 (see *line 442-445*).

Thanks so much.

Reviewer #2 (Remarks to the Author):

The submitted revision keeps the fundamental flaws of the original submission.

1, In authors' rebuttal, they stated "The RMSD of nucleoproteins between MeV and SeV is 1.4 Å", which means the two structures are identical given the experimental conditions. The Fig. 1b in this manuscript showed identical structure as that in Fig. 2A in Gutsche et al. 2015. There is no "newly discovered hole". No one can discover a new structure feature if the structure is identical to a previously published structure, period. The conformation of the truncated C-terminal region in this structure is artificial, having no relevance in the infectious nucleocapsid. In conclusion, this reported structure is identical to that of MeV nucleocapsid.

Re:

Thanks so much for the criticism.

We have followed the advice of Reviewer #1 and done sequence and structural comparison of nucleoproteins in the family of *Paramyxoviridae*. Nucleoproteins including SeV, NDV, MuV, NiV, PIV5 and MeV exhibit low sequence identity (~26%) but high structural conservation (RMSD < 1.4 Å) (*Supplementary Fig. 12*) (P2P_Figure 1). Surprisingly, these similar subunits assemble into different oligomeric states, including PIV5 ring structure, MeV helical structure and NDV/SeV clam-shaped structures under given experimental conditions. In our manuscript, double-headed SeV nucleocapsids derived from clam-shaped structures are what we want to focus.

We also performed analysis on three consecutive protomers from NDV clam-shaped structure, PIV5 ring structure and MeV helical structure. Similar to the interface between N-hole and Loop₂₄₀₋₂₄₈ in SeV, N-hole like structures and the paired extended loops also occur in the other three paramyxoviruses (*Supplementary Fig. 9*) (P2P_Figure 2), which indicates the popularity of the swapped N-hole in the assembly of nucleocapsids. We followed the reviewer's advice and changed "the newly discovered hole" to "unnoticed hole".

Thanks.

2, The authors continue to perpetuate biological relevance of "the clam-shaped structure". State here again: The genome of paramyxoviruses is a single stranded negative strand RNA, encapsidated in the nucleocapsid. A single strand genome in a virion has the full functions for virus replication. The authors continue to make erroneous statements: "the number of clam-shaped structures might be limited to 1 or 2 per virion", in Fig. 5, they claim "In one SeV nucleocapsid, clam-shaped joint and straight/condensed filaments coexist with loosed or even uncoiled filaments". Impossible!!! It is simple math, one nucleocapsid cannot have two heads, it is a single strand. The large portion of infectious SeV virions contains a single nucleocapsid and are fully infectious. "these double-headed nucleocapsid structures can confer benefits for genome stability, polyploid genome organization and genome condensation". A single nucleocapsid is completely stable, has normal condensation, and is fully infectious. If the authors believe that this pairing is relevant to multiple nucleocapsids in a virion, they need more data to support this claim, but it is not relevant to the infectivity of paramyxoviruses, especially SeV.

Re:

Thanks so much for the criticism.

Since the first discovery of clam-shaped structure in NDV nucleoprotein, we are curious to know the popularity and biological function of clam-shaped structures. Recently, similar

clam-shaped structures of nucleoproteins have been found in SeV and NiV from our lab and Antson's lab (*De-Sheng Ker, bioRxiv 2020*). Certainly, not all paramyxoviruses have been found to contain such structures and the occurrence of clam-shaped structures could be due to preparative techniques. The specific purification and cellular conditions in which this nucleocapsid type is preferentially formed is worth investigating. Meanwhile, the biological relevance is still unclear at this point and needs further studies.

We took the advice from the reviewer, and have toned down our voice on the biological relevance of the clam-shaped structures (see *line 224-249*).

Thanks so much.

Reviewer #3 (Remarks to the Author):

In the revised manuscript, the authors have addressed all questions posed by this referee. There are no further concerns and the manuscript may be accepted for publication.

Re:

Thanks so much for the great support.

REVIEWERS' COMMENTS:

Reviewer #1 (Remarks to the Author):

Thank you for your efforts and the modifications of the manuscript.

I would like to kindly ask you to consider the following changes:

A) Please mention the RMSD values of N of different paramyxoviruses in the introduction and also move the current supplementary figure 12 there.

B) Thank you for adding the figure highlighting the hole structure in different paramyxoviruses. I would find it helpful if you could comment in the text on the different charges observed in this structure from different paramyxoviruses and the implications.

C) Thank you for improving the tomography data. Unfortunately, I still have issues to recognise the clam shaped structures you are indicating. For readers that are not as familiar with this structure as you are, I would still consider it very helpful to segment the densities in amira or imod (or whichever program you would like to employ) to make visualisation easier for the reader.

Reviewer #2 (Remarks to the Author):

The manuscript is acceptable for publication.

Reviewers' comments:

Reviewer #1 (Remarks to the Author):

Thank you for your efforts and the modifications of the manuscript.

I would like to kindly ask you to consider the following changes:

A) Please mention the RMSD values of N of different paramyxoviruses in the introduction and also move the current supplementary figure 12 there.

Re:

Thanks so much for the great suggestion.

In *supplementary Figure 12*, we performed the structural comparison between SeV nucleoprotein protomer and other paramyxovirus nucleoproteins. We intended to follow the suggestion to move *supplementary Figure 12* to the *Introduction*. However, we noticed that “the *Introduction* should contain no references to figures or tables”, as listed in the AIP_checklist file. Meanwhile, it seems not perfect to mention SeV structures in the *Introduction*, before they are resolved in the following *Results*.

Based on these, we moved *supplementary Figure 12* to line 101 -104 as a new *supplementary Figure 5*. We also followed the suggestion and mentioned “*Despite of poor sequence conservation, paramyxovirus nucleoproteins exhibit well-conserved architectures*” in the *Introduction*.

Thanks again for the great suggestion.

B) Thank you for adding the figure highlighting the hole structure in different paramyxoviruses. I would find it helpful if you could comment in the text on the different charges observed in this structure from different paramyxoviruses and the implications.

Re:

Thanks so much for the fantastic suggestion.

We have emphasized the popularity of the hole structures in different paramyxovirus nucleoproteins, and added the sentences as “*Thus, these conserved interfaces between N-holes and the extended loops resemble a gate latch and bolt, and apparently function to tightly anchor the positions of neighboring nucleoprotein protomers. Therefore, N-hole adopts the same domain swapping process as N-arm and C-arm, and contributes to the assembly of helical nucleocapsids in the family of Paramyxoviridae.*” (See line 152-156).

Thanks.

C) Thank you for improving the tomography data. Unfortunately, I still have issues to recognise the clam shaped structures you are indicating. For readers that are not as familiar with this structure as you are, I would still consider it very helpful to segment the densities in

amira or imod (or whichever program you would like to employ) to make visualisation easier for the reader.

Re:

Thanks so much for the great suggestion.

We intended to do cryo-ET on intact SeV virions to verify the existence of double-headed SeV nucleocapsids. Unfortunately, the nucleocapsids wrap each other and it is quite hard to see structural details. We are still collecting more tomography data, and doing tomography averaging on intact SeV virions especially on those with larger size in diameter (>300 nm).

As an alternative, we turned to use Triton to lyse SeV virions to spread out nucleocapsids on cryo-EM grids. We collected cryo-EM micrographs instead of tomograms on the released nucleocapsids, and selected good candidates for clam-shaped structures. Thus, we cannot perform segmentation on the typical micrographs. Sorry for the confusion.

Thanks.

Reviewer #2 (Remarks to the Author):

The manuscript is acceptable for publication.

Re:

Thanks so much for the great support.